# Mechanisms of deformation and failure in colluvial slope under artificial surcharge loading

**Pei Zuan**⊙, **Jiali Feng**⊙⊙*, **Fenglin Ren**⊙

Chengdu University of Technology, Cheng Du, Sichuan, China

⊙ These authors contributed equally to this work.
* fengjiali0718@outlook.com

## Abstract

Large-scale artificial surcharge loading often triggers landslides in colluvial deposits, yet the mechanical response and failure mechanisms of such slopes under loading remain insufficiently understood. Using a typical colluvial slope in Tibet as a case study, this research integrates physical model tests and FLAC³ᴰ simulations to investigate loading-induced deformation and failure processes. Analogous materials composed of river sand, barite powder, calcium carbonate powder, and water were prepared, and multiple regression analysis was used to establish empirical relationships between mix ratios and the resulting cohesion and internal friction angle, yielding high similarity ($R_a^2 = 0.8732 - 0.9326$). Under loading, the slope exhibits maximum vertical and horizontal displacements of 40 mm and 50 mm, respectively, with shear stress concentrated along the loading boundary and vertical stress penetrating deeper than horizontal stress. The slope undergoes progressive failure: loading→initial equilibrium failure→rear-edge tensile cracking→upper soil mass sliding→front-edge extrusion and bulging→sliding surface propagation→overall failure. Furthermore, the colluvial slope exhibits pronounced failure sensitivity under loading, particularly in the progressive development of rear-edge tensile cracking, toe bulging, and deep shear bands, which should be regarded as key indicators for monitoring. These findings clarify the typical loading-induced failure mechanisms of accumulation-body slopes and provide a scientific basis for early landslide identification and hazard mitigation.

## 1. Introduction

Landslides are a common yet highly destructive form of geological hazard [1], often resulting in serious casualties, substantial property damage, and ecological degradation, thereby posing significant threats to regional economic and social development [2–4]. Statistical reports indicate that landslides generate approximately 20 billion USD in economic losses annually, representing 17% of the average yearly global

**Data availability statement:** All relevant data are within the manuscript and its Supporting Information files.

**Funding:** The author(s) received no specific funding for this work.

**Competing interests:** The authors have declared that no competing interests exist.

losses caused by natural disasters [5]. With ongoing population growth and the continued expansion of infrastructure construction into mountainous regions, both the frequency and the spatial extent of landslide hazards have progressively increased [6–8]. From 2004 to 2016, a total of 4,862 fatal landslides triggered by non-seismic factors were documented worldwide, resulting in 55,997 deaths [9]. Among the various types of landslides, those occurring within accumulation bodies are particularly prevalent due to their complex material compositions, loose structural characteristics, wide particle-size distributions, and poor cementation [10,11]. Such accumulation bodies commonly develop weak structural surfaces and highly permeable channels; under external influences, they exhibit heightened deformation sensitivity and a greater propensity for instability [12], leading to failure processes that are more abrupt and destructive [13–15].

Research on accumulation-body slopes has primarily focused on the influence of natural factors such as rainfall and earthquakes on their stability. Khan et al. [16] using physical analysis and finite-element numerical simulations, demonstrated the effects of rainfall infiltration on the stability of unsaturated coal-gangue accumulation-body slopes, and found that coupled stress-pore-water-pressure analysis provides more accurate predictions of slope stability than hydraulic analysis alone. Gan et al. [17] employing controlled dual-permeability artificial rainfall technology and dynamic sensor monitoring, systematically examined the effects of rainfall infiltration on the stability of accumulation-body slopes containing weak interlayers. Their findings showed that the deformation and failure processes could be divided into three stages: creep initiation, accumulation-uplift, and accelerated sliding. Manish et al. [18] through material characterization and scaled flume experiments, clarified the mechanism of rainfall -erosion -induced instability in accumulation-body slopes, revealing that failure mainly results from reductions in unsaturated shear strength caused by rainfall rather than sudden increases in pore-water pressure. Zhou et al. [19] integrating engineering-geological investigation, laboratory mechanical testing, and numerical simulation, identified the mechanism of large-scale landslide failure under rainfall infiltration. Their results indicated that rainfall infiltration, together with toe excavation, weakens the structural strength of the accumulation body and induces shallow sliding. Mohanty et al. [20] based on field monitoring and numerical analysis, revealed differences in the dynamic responses and stability of coal-mine overburden dump-slope accumulation bodies under seismic loading caused by material heterogeneity. Zhao et al. [21] and Li et al. [22] conducted numerical simulations of slope seismic responses using dynamic time-history analysis and evaluated slope seismic stability on this basis. Ma et al. [23] through shaking-table experiments on accumulation-body slopes, demonstrated that seismic loading leads to acceleration amplification with increasing slope height, progressive crack development, and ultimately bulldozing-type sliding failure.

As an important external load commonly encountered in mountainous transportation engineering, mining operations, and land-use development [24], surcharge loading is characterized by sudden application, concentrated distribution, and high intensity [25–27]. Such loading can substantially alter the stress and seepage fields

of slopes [28], accelerate the accumulation of local damage and the formation of slip surfaces, and consequently cause the failure mode and instability evolution to differ significantly from those under natural conditions [29]. However, existing studies provide insufficient attention to this issue, and systematic investigations into the deformation response, failure-process evolution, and controlling mechanisms of accumulation-body slopes under surcharge loading remain limited. This research gap is particularly pronounced in plateau mountainous regions, where large topographic relief and complex rock-soil structures further complicate slope behavior.

To address these gaps, this study investigates a typical deposit slope in Tibet, integrating laboratory physical model experiments with numerical simulations to systematically examine slope deformation and failure characteristics under external loading, while revealing the underlying instability mechanisms. The results provide a theoretical foundation for the siting of major infrastructure and slope-support design, as well as technical guidance for enhancing early detection and risk management of deposit slope failures.

## 1.1. Research subject

The investigated deposit slope is located in the southeastern part of the Tibet Autonomous Region, China, within a high-altitude alluvial-lacustrine terrace landform. Long-term coupled erosion and weathering have caused pronounced bank slope degradation. Local channel widening has led to continuous expansion toward both banks, while persistent fluvial scouring at the slope toe has produced a serrated terrain with steep inclinations ranging from 21° to 93° (Figs 1). The slope extends about 450 m laterally, 185 m longitudinally, and covers a planar area of approximately 83,250 m², presenting a typical bench-like structure (Fig 1b). At the rear edge, a 10 m-high collapse scarp with an 80° inclination is accompanied by NW60° unloading cracks (Fig 1a). The central portion of the landslide body is densely segmented by step-like scarps, the largest offset reaching about 4.0 m (Fig 1d). At the toe, cracks in the protective dike reach widths of up to 20 cm (Fig 1c). The primary external loading on the slope arises from the deposition of construction waste, which, combined with the slope's self-weight, significantly increases the likelihood of landslide initiation. Field investigations and

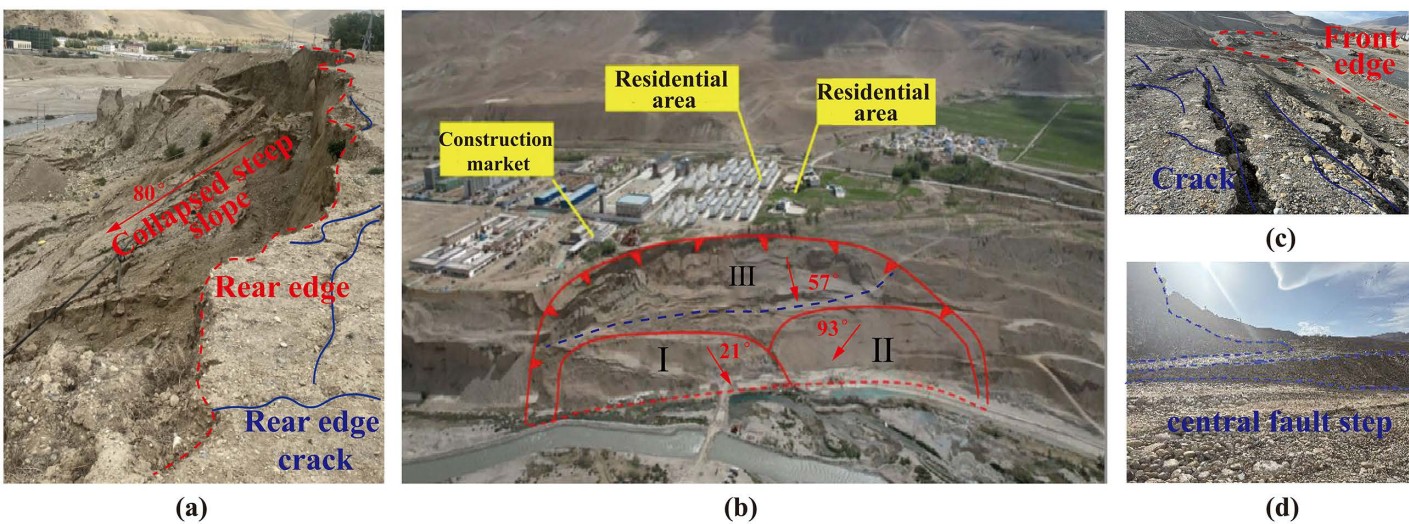

**Fig 1. Field characteristics of the deposit slope:** (a) rear-edge collapse scarp and unloading cracks; (b) aerial view of the entire slope with zoning; (c) toe boundary and surface cracks; (d) densely developed step-like scarps in the central portion. All photographs were taken by the authors during field investigations and are original works. The authors hold full copyright and agree to publish the images under the Creative Commons Attribution (CC BY 4.0) license.

borehole data indicate that the slope has an average thickness of 10.5 m and is mainly composed of glaciofluvial gravelly soil and colluvial gravelly soil.

The stratigraphic sequence of the slope, from youngest to oldest, consists of: Quaternary Holocene artificial fill ($Q_4^{ml}$), Holocene cultivated soil ($Q_4^{pd}$), Holocene landslide deposits($Q_4^{del}$), Holocene colluvial-alluvial deposits ($Q_4^{al+pl}$), and Upper Pleistocene glaciofluvial deposits ($Q_3^{fgl}$). Based on the results of the geotechnical survey, the slope is primarily composed of three strata: $Q_4^{del}$, $Q_4^{al+pl}$, and $Q_3^{fgl}$. The main materials include cobble-bearing soil, cultivated soil, and plain fill. The engineering geological cross-section is shown in Fig 2.

## 2. Research methodology

The primary research methods adopted in this study include laboratory model tests and numerical simulations. First, based on the data obtained from field investigations and geological surveys, similar materials were prepared to conduct physical model tests in the laboratory. Because the actual loading conditions of the slope in the Xiangchang area could not be directly obtained, and considering the limitations of the physical model tests, the loads applied in the numerical simulations were converted from the loading values used in the physical model tests according to similarity ratios, whereas the geometric dimensions were kept consistent with the field conditions. The combination of these two approaches enables comparative validation and further clarification of the deformation and failure mechanisms of the colluvial slope under artificial stacking loads. The workflow is illustrated in Fig 3.

### 2.1. Physical model test

Due to the high cost, long duration, and limited controllability of field experiments [30], as well as the difficulty of reproducing specific environmental conditions, this study employs laboratory-scale physical model experiments, which offer high controllability, reproducibility, and relatively low cost [31]. The experimental procedure is illustrated in Fig 4.

**2.1.1. Similarity material proportioning.** The development of analogous materials is a prerequisite for conducting physical model experiments and is a critical factor in determining whether the model can accurately replicate the physical

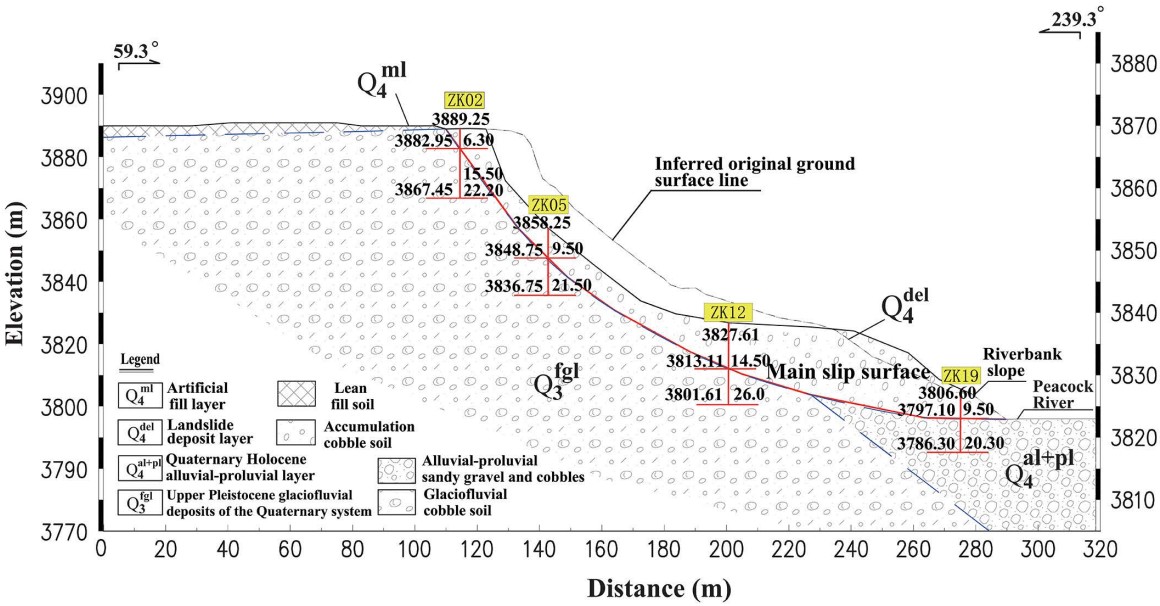

**Fig 2. Engineering geological profile of the slope.**

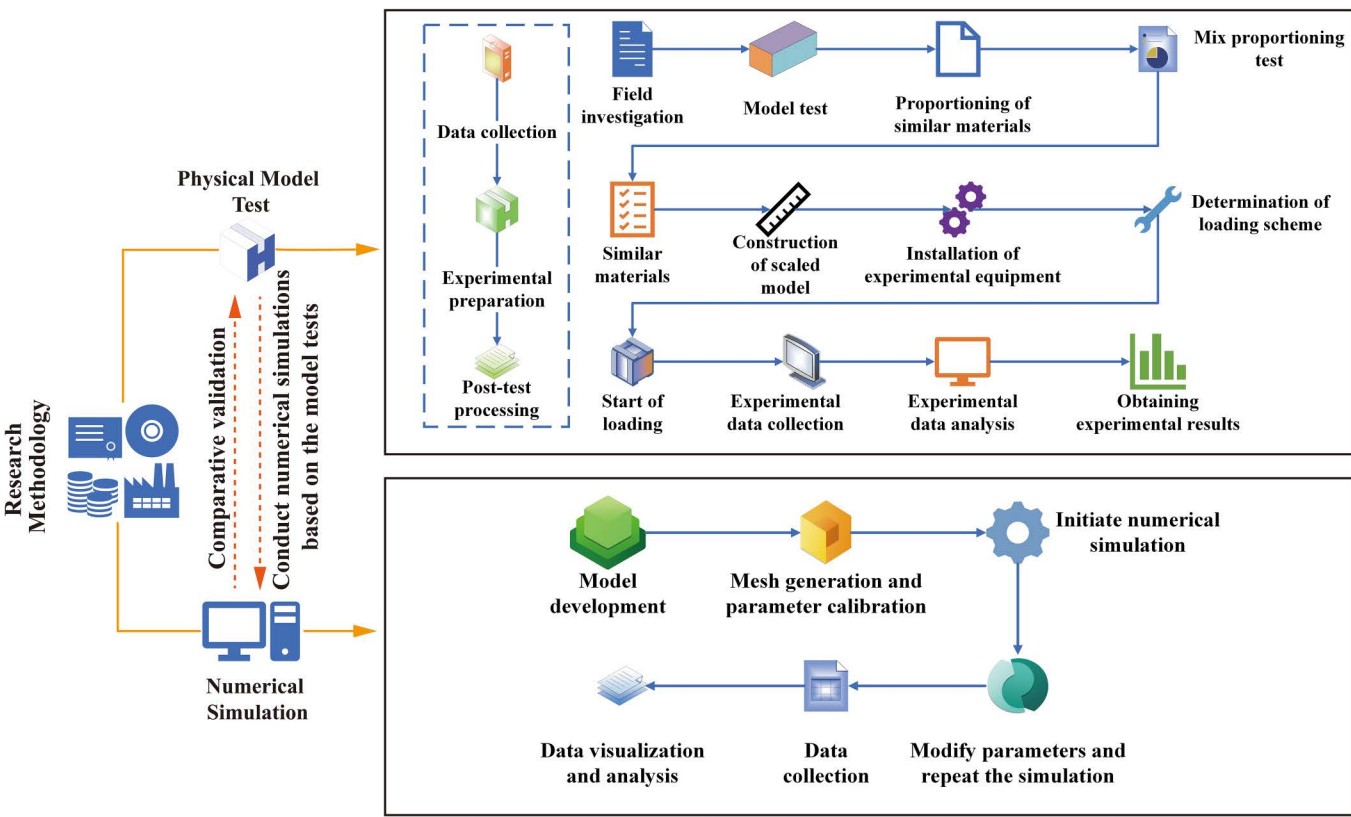

**Fig 3. Overall workflow of the research methodology.**

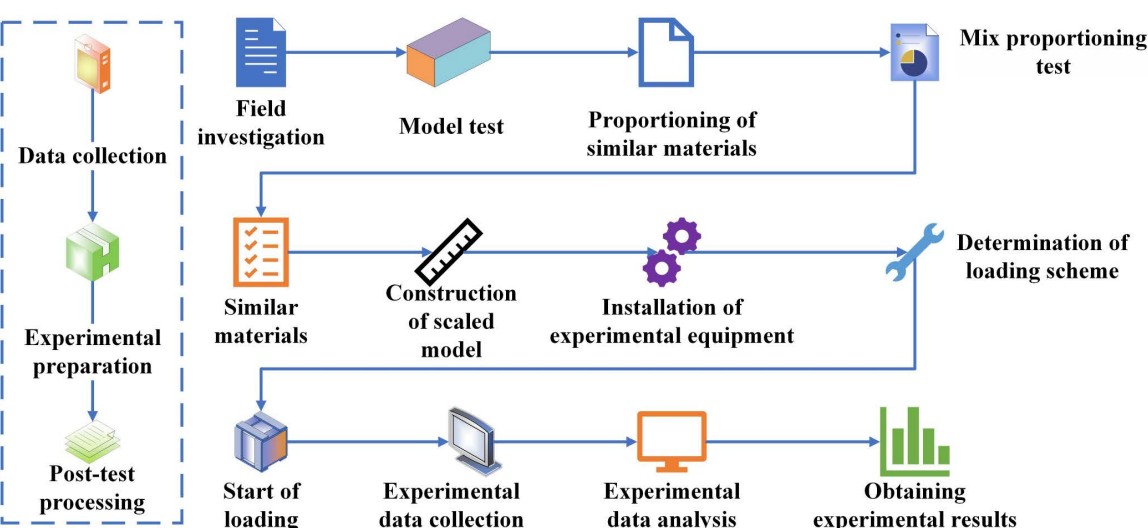

**Fig 4. Experimental procedure flowchart.**

properties of the prototype [32]. Based on similarity theory, this study establishes quantitative relationships between the model and the prototype to reveal the underlying mechanisms of complex phenomena. To ensure that the model's deformation behavior corresponds to that of the actual engineering system, the model must satisfy the required similarity criteria. All relevant parameters can be represented by the following relationship, as shown in Eq. 1.

$$f = (l, \gamma, g, c, \varphi, E, \delta, \sigma, \varepsilon, \mu, t, F) = 0 \tag{1}$$

Where, $l$ denotes the model size, $\gamma$ is the unit weight of the model, $g$ represents gravitational acceleration, $c$ is cohesion, $\varphi$ is the internal friction angle, $E$ is the elastic modulus of the model, $\delta$ is displacement, $\sigma$ is stress, $\varepsilon$ is strain, $\mu$ is Poisson's ratio, $t$ is time, and $F$ is force.

Because $\varphi$, $\varepsilon$, and $\mu$ are dimensionless quantities, their similarity ratios are equal to 1, that is, $C_\varphi = C_\mu = C_\varepsilon = 1$. When $C\gamma$ is set to 1, the derivation of the remaining physical quantities must satisfy the similarity criteria specified in Eqs. 2–4.

$$C_l = l_p / l_m \tag{2}$$

Where, $C_l$ is the geometric similarity constant with the dimension of length, where the subscript $p$ denotes the prototype and the subscript m denotes the model.

$$\begin{cases} C_\sigma = C_l C_\gamma \\ C_\sigma = C_E C_\varepsilon \\ C_\delta = C_l \\ C_F = C_\sigma C_l^2 \end{cases} \tag{3}$$

Where, $C_\sigma$ denotes the stress similarity constant, $C_\gamma$ represents the unit-weight similarity constant, $C_E$ is the elastic modulus similarity constant, $C_\varepsilon$ is the strain similarity constant, $C_\delta$ is the displacement similarity constant, and $C_F$ is the force similarity constant.

$$\begin{cases} C_\sigma = C_R^\tau \\ C_c = C_R^\tau \\ C_R^c = C_R^\tau \end{cases} \tag{4}$$

Where, $C_R^\tau$ denotes the tensile strength similarity constant, $C_c$ represents the cohesion similarity constant, and $C_R^c$ is the compressive strength similarity constant.

The principal geometric dimensions of the debris-slope cross-section are shown in Fig 5. To ensure experimental operability, the geometric similarity constant was set to 200 in the model tests, and the similarity ratios for the other physical quantities are provided in Table 1.

The soil's mechanical parameters were determined through laboratory testing, as summarized in Table 2. The testing procedure is shown in Fig 6. The natural density of the soil was measured using the cutting-ring method, while the maximum dry density and optimum moisture content were obtained from compaction tests. Cohesion, internal friction angle, deformation modulus, and Poisson's ratio were derived from large-scale triaxial tests. Detailed testing procedures are provided in Appendix A.

Based on the properties of in-situ soils and relevant literature [33,34], and considering the particle size effects in model experiments, the analogous materials were required to be silty and granular, with low cohesion and moderate-to-low strength. We reviewed the mix proportions of soil-simulating materials reported in comparable studies, as summarized in Table 3.

As shown in Table 3, the mix materials that recur across various studies are primarily gypsum, barite powder, and quartz sand. Existing research indicates that barite powder is mainly used to adjust material density, quartz sand provides

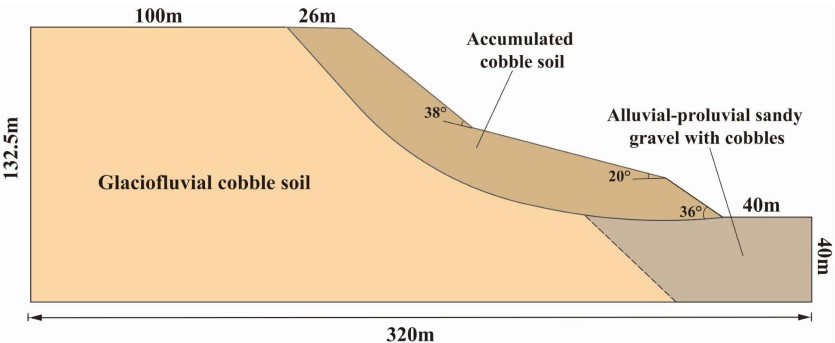

**Fig 5. Geometric dimensions of the debris-slope cross-section.**

**Table 1. Similarity parameters between model and prototype.**

| Physical Quantity | Dimension | Similar Quantity | Similarity Ratio |
|---|---|---|---|
| Geometric length ($L$) | L | $C_L$ | 200 |
| Internal friction ($\phi$) | 1 | $C_\phi$ | 1 |
| Cohesion ($c$) | $FL^{-1}T^{-2}$ | $C_c$ | 200 |
| Water content ($\omega$) | 1 | $C_\omega$ | 1 |
| Unit weight ($\gamma$) | $FL^{-3}$ | $C_\gamma$ | 1 |
| Strain ($\varepsilon$) | 1 | $C_\varepsilon$ | 1 |
| Elastic modulus ($E$) | $FL^{-2}$ | $C_E$ | 200 |
| Poisson's ratio ($\mu$) | 1 | $C_\mu$ | 1 |
| Displacement ($\delta$) | L | $C_\delta$ | 200 |
| Stress ($\sigma$) | $FL^{-2}$ | $C_\sigma$ | 200 |
| Force ($F$) | F | $C_F$ | $200^3$ |
| Flexural rigidity ($EI$) | $FL^2$ | $C_{EI}$ | $200^5$ |
| Time ($t$) | T | $C_t$ | $200^{1/2}$ |

**Table 2. Main mechanical parameters of the slope soil.**

| Stratum | Soil Type | Natural Density (g/cm³) | Deformation Modulus (MPa) | Poisson's Ratioμ | Cohesion (kPa) | Internal Friction Angle $\varphi$ (°) |
|---|---|---|---|---|---|---|
| $Q_4^{del}$ | Accumulated gravelly soil | 2.02 | 24.5 | 0.3 | 8.7 | 31.0 |
| $Q_3^{fgl}$ | Fluvio-glacial gravelly sand | 2.07 | 24.5 | 0.3 | 9.3 | 32.1 |
| $Q_4^{al+pl}$ | Ice-water accumulated gravelly soil | 2.09 | 24 | 0.3 | 8.8 | 29.8 |

a granular framework or modifies the friction angle, and gypsum, as a commonly used cementing agent, enhances cohesion and overall strength. However, since the similar material in this study must exhibit powder-like particle characteristics, gypsum is not suitable as a cementing component. A review of the relevant literature [41–44] shows that double fly ash powder is inexpensive and exhibits non-hardening and non-hydrating properties, making it well suited as an inert filler in weakly cemented or powder-soil similar materials. Therefore, considering material accuracy, controllability, and economic feasibility, the final mix system selected in this study consists of barite powder, river sand, double fly ash powder, and water.

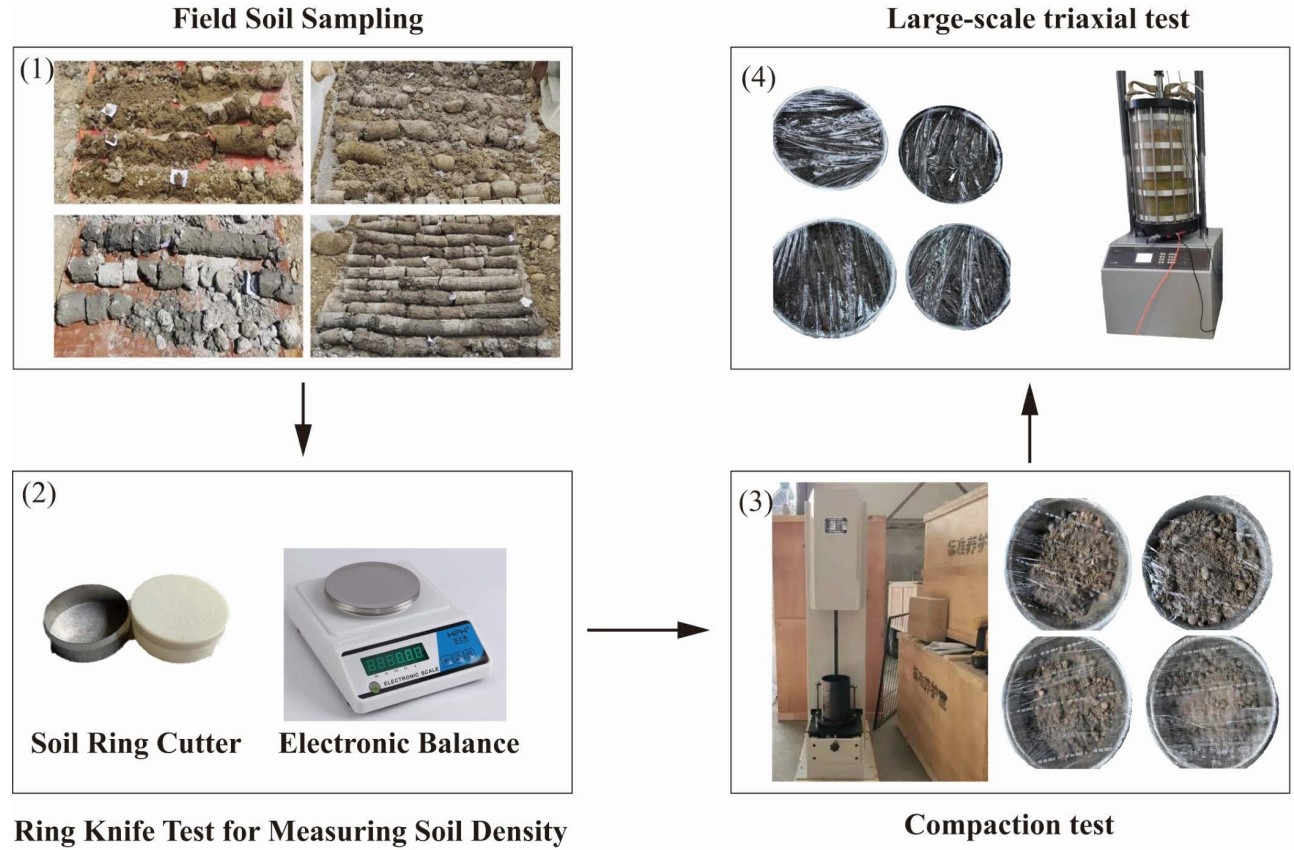

**Fig 6. Procedure for obtaining soil mechanical parameters.**

**Table 3. Mix Proportions of Soil-Simulating Materials in Comparable Studies.**

| Serial Number | Analog Material System | Simulation Type | Source |
|---|---|---|---|
| 1 | Gypsum + Barite Powder + Bentonite + Sand | Accumulation with Weak Interlayers | [35] |
| 2 | Gypsum + River Sand + Barite + Cementing Agent | Layered Accumulation, Fractured Rock Mass and Stratified Slope | [36] |
| 3 | Barite Powder + Gypsum + Quartz Sand + Water | Stratified Slope, Sedimentary Sand Layer and Non-Cohesive Depositional Soil | [37] |
| 4 | Gypsum + Fly Ash/Iron Powder + Barite + Loose Sand | Large-Deformation Accumulation and Weathered Material Accumulation | [38] |
| 5 | River Sand + Barite + Gypsum + Cementing Agent | Generic Accumulation Simulation | [32] |
| 6 | Standard Sand + Barite Powder + Bentonite + Accumulation Soil | Man-Made Accumulation Slope | [39] |
| 7 | Barite Powder + Quartz Sand + Gypsum + Glycerol + Water | Conventional Rock Slope | [40] |

The experimental materials consisted of river sand, barite powder, double fly ash powder, and water, with their properties listed in Table 4. River sand served as the skeleton material, barite powder was used to adjust the model density, and double fly ash powder acted as fine filler to occupy voids and reduce cohesion.

The proportioning experiments for the analogous materials were conducted using the orthogonal design method [45–48]. This method allows for the selection of uniformly distributed and representative schemes from all possible combinations of experiments while considering multiple influencing factors, thereby effectively reducing the number of experimental groups and efficiently analyzing the effects of each factor on the results [49]. In the proportioning experiments, three factors were considered: the mass ratio of river sand to aggregate (factor A), barite powder to aggregate (factor B), and double fly ash powder to aggregate (factor C), each with five levels, resulting in a total of 15 experimental groups. To facilitate comparative analysis of the effects of different factors and to eliminate the influence of irrelevant conditions on the results, the moisture content of all analogous materials was fixed at 10% of the aggregate mass. The orthogonal design levels of the analogous materials are listed in Table 5.

Based on the orthogonal experimental design, the raw materials were prepared, samples were fabricated, and relevant parameters were tested. The experimental procedure is illustrated in Fig 7.

The proportions and physical properties of the 15 sets of prepared analogous materials are presented in Table 6. Studies have shown that the shear strength parameters of geotechnical materials—cohesion $c$ and internal friction angle $\varphi$ are key factors governing the slope safety factor. These parameters directly influence both the occurrence of slope failure and the potential failure mode, and thus should be treated as primary variables during material proportioning [50–53]. The results indicate that, within the selected proportioning range, the density ranged from 1.953 to 2.098 g/cm³, cohesion from 8.66 to 9.19 kPa, and the internal friction angle from 29.7° to 32.4°.

**2.1.2. Influence mechanisms of various factors on similar materials.** This study employed the range analysis method to examine how the four component factors A, B and C influence the material's physical-mechanical parameters and hydraulic properties. The range is calculated as the difference between the maximum and minimum average values across the levels of each factor. A larger range indicates that the corresponding factor exerts a stronger influence on the properties of the similar materials.

**2.1.2.1. Analysis of factors affecting the density of similar materials:** Based on the orthogonal test results, a range analysis was performed to evaluate the density of the similar materials. The average values for each level of factors A–C were calculated, and the corresponding line chart was plotted, as shown in Fig 8. The results indicate that the relative

**Table 4. Material parameters of the similarity model components.**

| Raw material | Appearance | Main component | Density/(g·m⁻³) |
|---|---|---|---|
| River sand | Light yellow grains | Quartz>90% | 2.65 |
| Baryte powder | White powder | Barium sulfate>95% | 4.3 |
| Double fly powder(Ultrafine calcium carbonate powder) | White powder | Calcium carbonate>90% | 2.8 |

The data in this table was supplied by the vendor.

**Table 5. Orthogonal design levels for similarity material composition.**

| Level | Factor A/% | Factor B/% | Factor C/% |
|---|---|---|---|
| 1 | 50 | 30 | 20 |
| 2 | 55 | 25 | 20 |
| 3 | 60 | 25 | 15 |
| 4 | 65 | 20 | 15 |
| 5 | 70 | 20 | 10 |

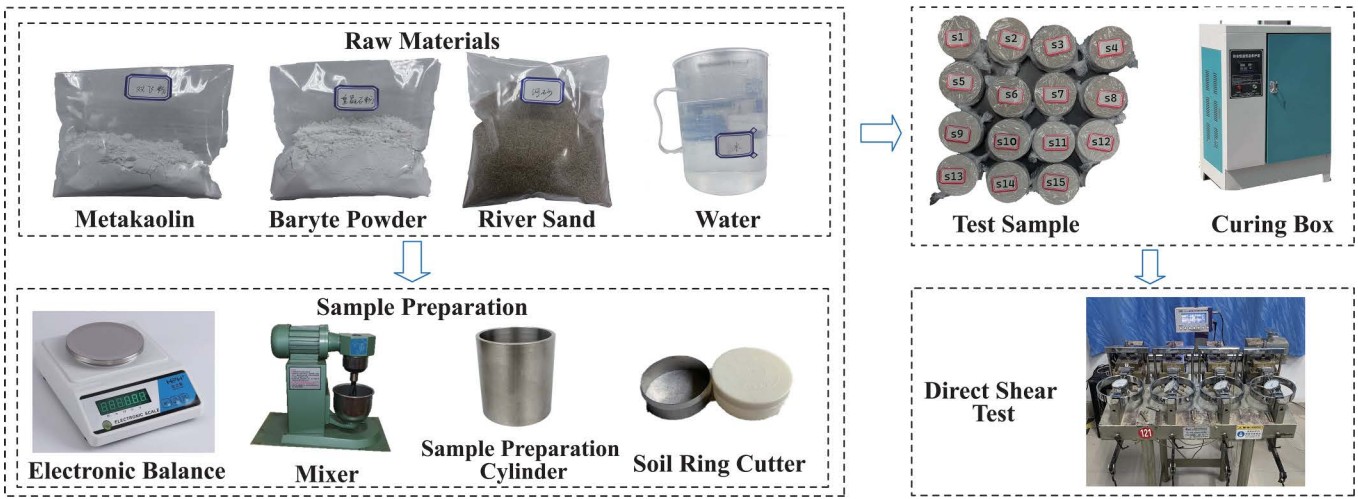

**Fig 7. Experimental procedure for preparing and testing similarity materials.**

**Table 6. Results of similarity material mix tests. The numbers in the mix schemes represent the orthogonal design levels. Factors A, B, and C correspond to the mass ratios at each level.**

| Test Number | Mixing scheme | Density (g/cm³) | Cohesion (kPa) | Internal friction angle (°) |
|---|---|---|---|---|
| S1 | 1: 1: 1 | 2.034 | 9.02 | 30.8 |
| S2 | 1: 2: 2 | 1.978 | 8.63 | 31.6 |
| S3 | 1: 3: 3 | 2.057 | 9.47 | 30.1 |
| S4 | 1: 4: 4 | 2.006 | 8.66 | 30.3 |
| S5 | 1: 5: 5 | 2.081 | 8.89 | 29.5 |
| S6 | 2: 1: 2 | 2.058 | 9.09 | 32.0 |
| S7 | 2: 2: 3 | 2.098 | 9.41 | 30.7 |
| S8 | 2: 3: 4 | 1.967 | 8.84 | 31.0 |
| S9 | 2: 4: 5 | 2.002 | 9.22 | 30.4 |
| S10 | 3: 1: 3 | 1.953 | 8.72 | 31.8 |
| S11 | 3: 2: 4 | 1.972 | 8.93 | 29.9 |
| S12 | 3: 3: 2 | 2.033 | 9.35 | 30.2 |
| S13 | 4: 1: 4 | 2.016 | 8.97 | 32.4 |
| S14 | 4: 2: 5 | 2.071 | 8.60 | 29.7 |
| S15 | 5: 1: 5 | 2.008 | 9.19 | 31.2 |

influence of the factors on the density of the similar materials decreases in the order of B (range: 0.077, with subsequent values also representing ranges)> A (0.0575)> and C (0.0503). Factor B (barite powder/aggregate) is the primary factor controlling the density. A lower barite powder content combined with higher contents of river sand and heavy calcium powder leads to an increase in the density of the similar materials. This finding further confirms that incorporating river sand and heavy calcium powder into the aggregate can effectively regulate the unit weight of similar materials.

**2.1.2.2. Analysis of factors affecting the cohesion and internal friction angle of similar materials:** Based on the test results for each group of similar materials, the average cohesion and internal friction angle corresponding to each factor were calculated under both natural and water-saturated conditions, and the resulting line chart is presented in Fig 9.

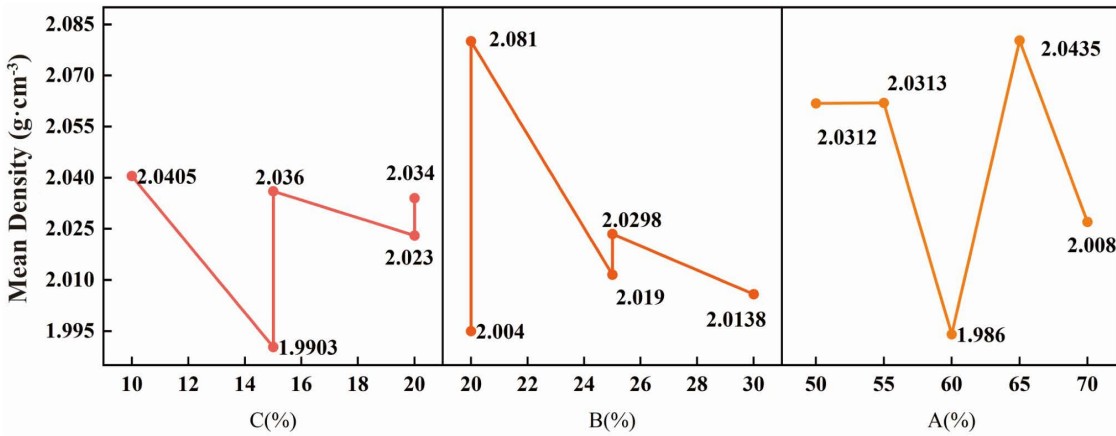

**Fig 8. Analysis of density sensitivity factors.**

Based on the calculations, the influence of each factor on cohesion is ranked as A (0.405) > C (0.350) > B (0.330). This result indicates that the sensitivity of cohesion is primarily governed by factor A (river sand/aggregate). As the river sand content increases, the interlocking and contact characteristics among the particles of the similar materials are altered, promoting the formation of a more stable internal structural framework that enhances cohesion. Although double fly ash powder and barite powder also exert an influence on cohesion, their effects are weaker compared with the changes induced by variations in aggregate particle gradation. These findings suggest that the cohesion of similar materials is pre-dominantly controlled by the configuration of the aggregate skeleton, while fine-grained materials mainly serve as fillers and provide localized cementation.

Similarly, the influence of each factor on the internal friction angle is ranked as B (2.14) > C (1.07) > A (0.74). Among these factors, factor B (barite powder/aggregate) exerts the most significant and dominant control over the internal friction angle. Variations in barite powder content lead to substantial changes in particle morphology, compactness, and inter-particle friction within the similar materials, resulting in pronounced differences in their frictional behavior. Barite powder is characterized by its high density and relatively smooth, low-angularity particles; therefore, when its content decreases, the rougher particle surfaces formed by river sand and double fly ash powder, together with increased aggregate contact strength, contribute to an increase in the internal friction angle. It is also notable that factor C (double fly ash powder/aggregate) has a measurable influence on the internal friction angle, primarily because double fly ash powder fills the voids within the skeleton and adjusts particle packing, thereby indirectly modifying the material's overall shear-resistance system.

In summary, cohesion is primarily governed by the structural characteristics of the aggregate particle framework, whereas the internal friction angle is mainly controlled by interparticle frictional resistance and material compactness. The differing sensitivities of these two strength parameters to the various factors reflect the distinct dominant mechanisms associated with internal structure and particle gradation during the shear behavior of similar materials. These results pro-vide a reliable basis for optimizing mixture proportions and conducting subsequent model tests involving similar materials.

**2.1.3. Multivariate regression analysis of parameters and determination of experimental material mix proportions.** In this study, multivariate regression analysis was conducted to quantify the relationships between each physical and mechanical parameter and the compositional proportions of the three factors A, B, and C. The fitting results for each parameter are presented in Fig 10. The coefficients of determination between the regression-derived fitted values and the corresponding experimental values range from $R_a^2 = 0.8732$ to $0.9326$, indicating a strong goodness of fit. The regression equations are provided in Eq. 5.

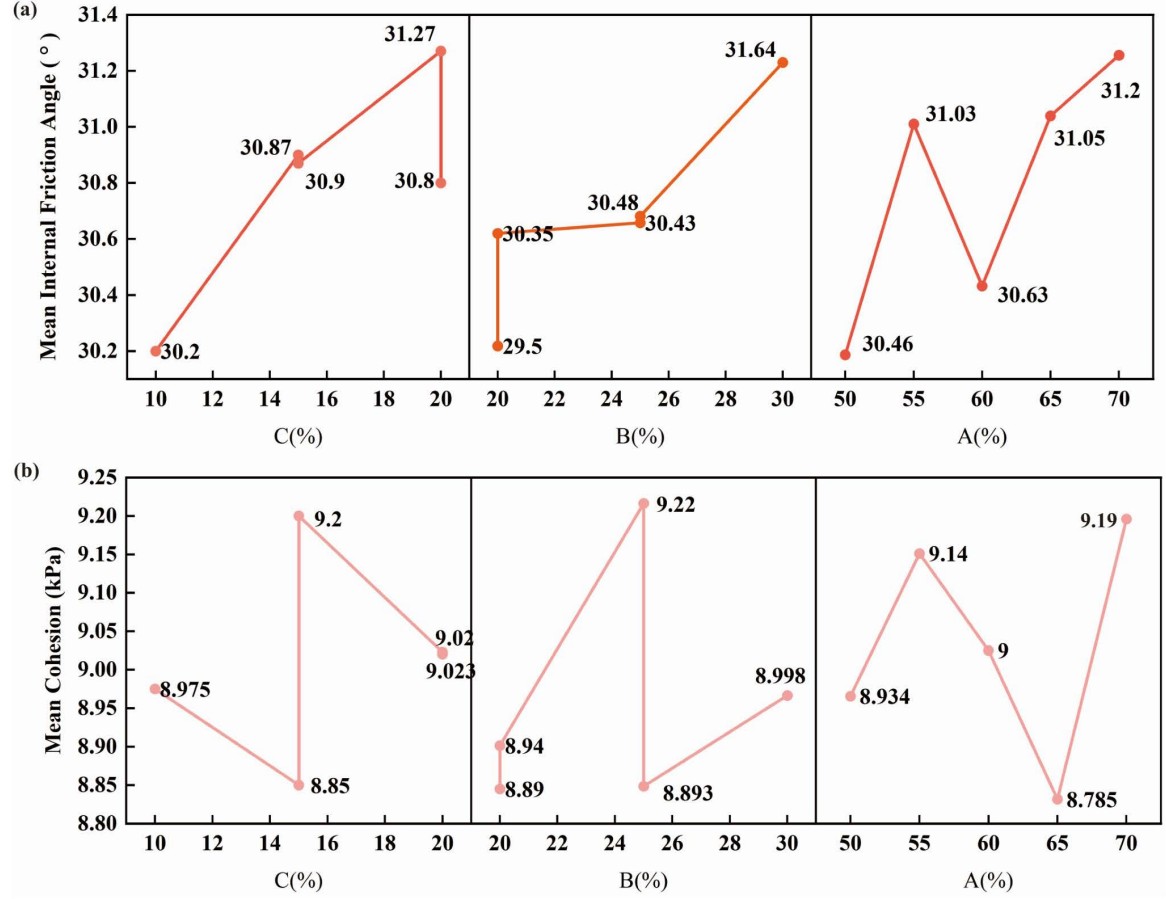

**Fig 9. Analysis of sensitivity factors affecting cohesion and internal friction angle.**

$$\begin{cases} \rho = 2.149 + 0.00191a - 0.00109b + 0.00308c \\ c = 8.857 + 0.00049a - 0.00515b + 0.00252c \\ \varphi = 26.576 - 0.00674a + 0.16055b + 0.03058c \end{cases}$$

(5)

Where, $a$ represents the percentage of river sand mass relative to the aggregate mass; $b$ represents the percentage of barite powder mass relative to the aggregate mass; and $c$ represents the percentage of double-fly ash powder mass relative to the aggregate mass.

Considering the relatively high variability inherent in analogous experiments [54], the optimal proportion of analogous materials was determined based on the physical and mechanical properties of the model materials. By combining the multiple regression equations of the factors with the results of sensitivity and correlation analyses, the optimal proportion for the model tests was calculated using Eq. 6.

$$w_{nm} = \sum_{i=1}^{9} kn \left| \frac{y_n - C_n y_{nm}}{y_n} \right|$$

(6)

Where, $y_n$ represents the value of the nth soil property; $y_{nm}$ is the value of the nth property for the $m$-th analogous material; $C_n$ denotes the similarity coefficient for the nth property; $k_n$ is the weight derived from the sensitivity analysis; and $w_{nm}$

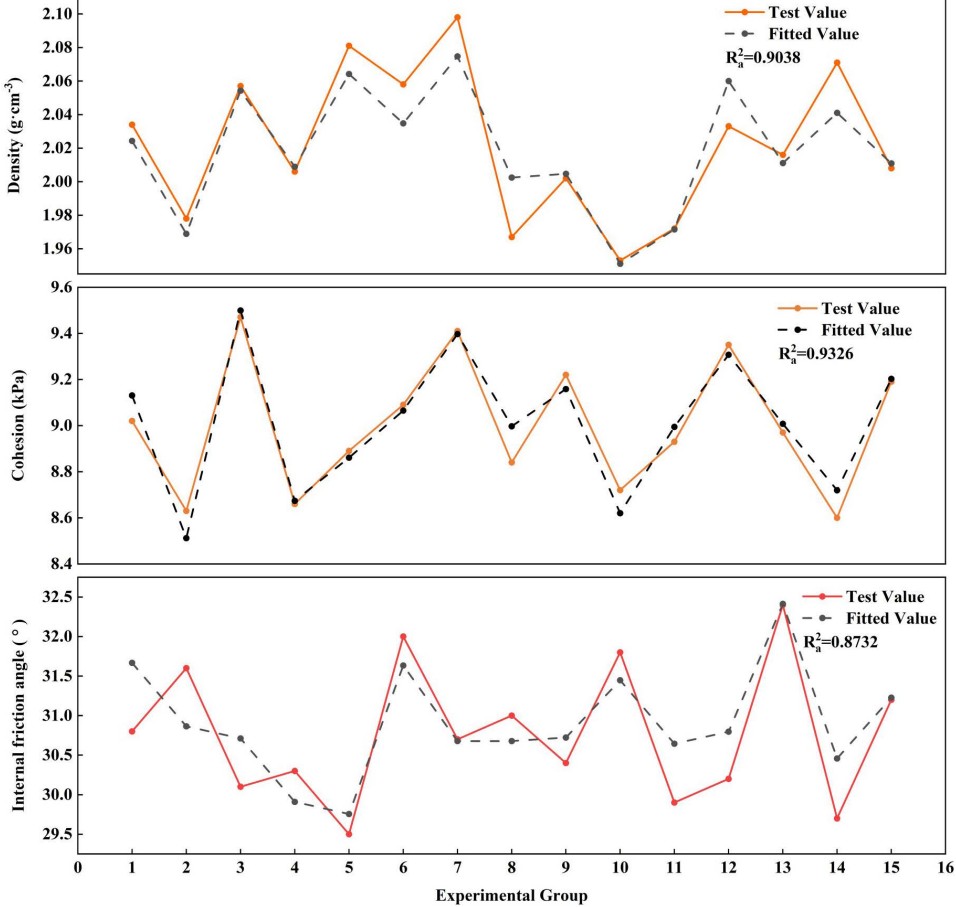

**Fig 10. Comparison of the multivariate regression fitting results with the experimental results for each parameter.**

represents the error between the fitted value of the analogous material with a given proportion and the actual value of the soil. The proportion corresponding to the minimum $w_{nm}$ is defined as the optimal proportion of the analogous material.

The final proportions of the analogous materials and their corresponding mechanical properties are summarized in Table 7.

**2.1.4. Physical model setup and experimental equipment.** The model was constructed based on the measured engineering geological profile, and a scaled-down slope model of the main region was built according to a 1:200 similarity criterion. A custom-designed model box with dimensions of 1.6 m × 1 m × 1.6 m was used for the experiments, as shown in

**Table 7. Optimal mixture ratios of similarity materials.**

| Category | River sand | Barite powder | Double fly powder | Water | Density (g/cm³) | Cohesion (kPa) | Internal friction angle (°) |
|---|---|---|---|---|---|---|---|
| $Q_4^{del}$ | 1 | 2.18 | 1.36 | 0.45 | 2.025 | 8.61 | 30.72 |
| $Q_3^{fgl}$ | 1 | 1.92 | 1.25 | 0.42 | 2.063 | 9.33 | 32.53 |
| $Q_4^{al+pl}$ | 1 | 1.86 | 1.90 | 0.48 | 2.087 | 8.79 | 29.66 |

Fig 11a. To ensure uniform distribution of the surcharge on the slope, 1-meter-long strip sandbags were employed to apply the load. The final front view of the constructed model is shown in Fig 11b.

To analyze the displacement and stress variations of the slope under surcharge conditions, displacement sensors, earth pressure sensors, and internal displacement monitoring points were installed at different locations of the slope. In addition, imaging equipment was set up to observe surface changes, as shown in Fig 12.

The monitoring layout was implemented as follows: (1) Displacement monitoring was conducted along the slope crest-mid-slope-toe profile using three sets of displacement gauges (Points 1–3) to capture the full-section displacement response throughout the entire surcharge process. (2) The dynamic internal displacement of the slope was measured using a displacement marker method. Monitoring points were discretely distributed within a 7×7 cm grid reference system outside the model box, and vertical and horizontal displacements after loading were quantitatively tracked by recording coordinates before and after loading. (3) Differentiated monitoring points were set in three typical regions, three points per region: the front edge of the surcharge area (A1-A3), the front edge to the central surcharge area (B1-B3), and the central surcharge area (C1-C3), to observe the spatiotemporal evolution of internal slope displacements under surcharge conditions. (4) Earth pressure sensors were used to measure vertical stress changes, with Sensors 1–4 installed sequentially from the slope crest into the interior, and Sensor 5 positioned at the slope toe. (5) Surface changes were primarily recorded using imaging equipment such as cameras.

The experimental monitoring equipment is shown in Fig 13. Monitoring Points 1−3 were measured using dial gauges, with data recorded every 30 minutes. A 7×7 cm grid was affixed to the exterior of the model box to establish discretely distributed monitoring points; by documenting the coordinates before and after loading, the vertical and horizontal displacement patterns were quantitatively tracked, with measurements taken every 30 minutes. Surface features were recorded using a Leica M10-R camera, with images captured once every 30 minutes. Earth pressure was measured using soil pressure sensors connected to a data acquisition instrument. The soil pressure sensor used was model SW-TYJ-20, and the data acquisition instrument, manufactured by Donghua Testing Corporation, was model DH3818−2, with a sampling frequency of one reading every 10 seconds.

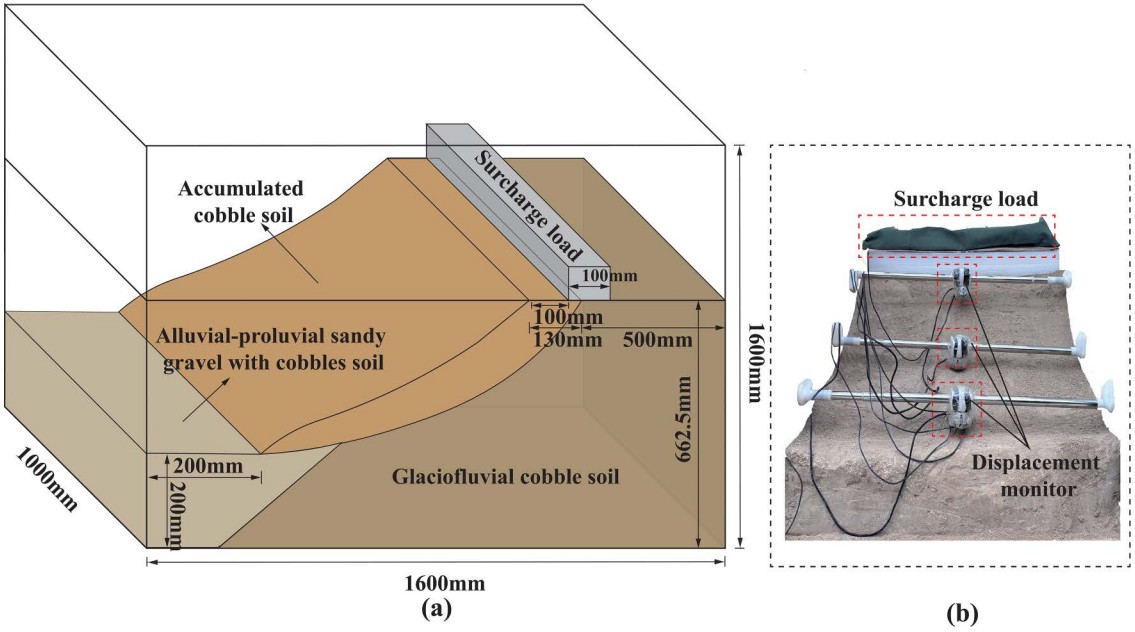

**Fig 11. Slope model design: (a) 3D design view; (b) frontal view of the physical model.**

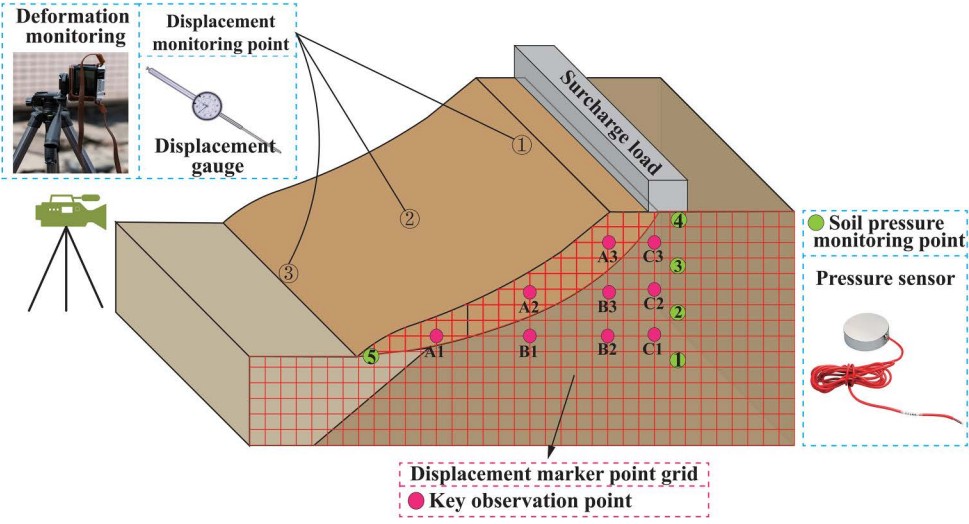

**Fig 12. Schematic layout of displacement and stress monitoring points in the slope.**

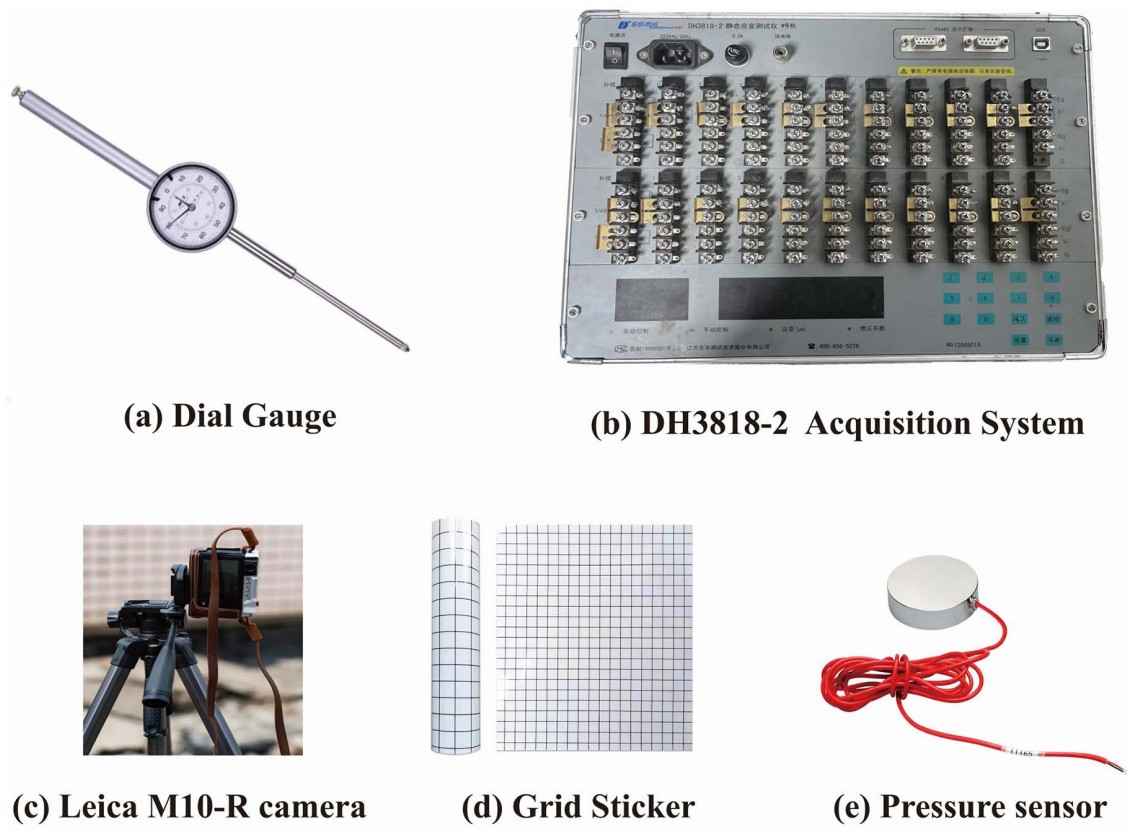

**(a) Dial Gauge**

**(b) DH3818-2 Acquisition System**

**(c) Leica M10-R camera**

**(d) Grid Sticker**

**(e) Pressure sensor**

**Fig 13. Experimental Equipment.**

**2.1.5. Artificial surcharge loading application scheme.** Based on the actual engineering conditions (the fill scale in the study area), the model testing specifications, and the experimental objectives (with emphasis on the progressive failure process of the slope), a staged loading scheme was adopted. Each load increment was set to 0.2 kPa, with a total of 10 load levels applied progressively, reaching a maximum load of 2.0 kPa. After applying each load level, a 2-hour holding period was maintained to ensure that slope deformation and stability fully developed under the current load, facilitating accurate capture of the slope's progressive failure. The loading-time curve is shown in Fig 14.

## 2.2. Numerical simulation

**2.2.1. Overall model configuration.** Due to inherent limitations of indoor scaled-down physical model tests, scale effects may cause discrepancies in mechanical responses compared to the prototype. The particle gradation, structural characteristics, and moisture conditions of the model soil cannot fully replicate those of the prototype, and the resolution and placement of sensors further restrict the precision in capturing subtle deformation and stress evolution [55]. Therefore, FLAC³ᴰ was employed to construct a numerical model with boundary conditions and physical parameters approximating the prototype, enabling the simulation of the slope's physical response under surcharge and the validation of experimental results.

The geological model was built according to the actual dimensions and morphology of the landslide, with a total length of 320 m, width of 450 m, and height of 132.5 m (Fig 15). The initial condition considered only the self-weight. Lateral boundaries were subjected to unidirectional displacement constraints, the bottom boundary to triaxial displacement constraints, and the natural slope surface was treated as a free boundary. The numerical calculation employed an elastoplastic constitutive model with the Mohr-Coulomb yield criterion.

**2.2.2. Determination of Soil Parameters through Inversion and Load Parameter Specification.** Appropriate selection of the physical and mechanical parameters of slope soils is crucial for numerical analysis. This ensures that the simulation accurately reproduces the deformation and failure mechanisms of the physical model and reliably predicts stability variations in the prototype slope. During the site investigation, both in-situ and laboratory physical-mechanical tests were conducted on various types of rock and soil masses. In total, six groups of sliding mass soil, six groups of slip zone soil, and twelve groups of sliding bed soil—including six groups of fluvial-alluvial pebble soil and six groups

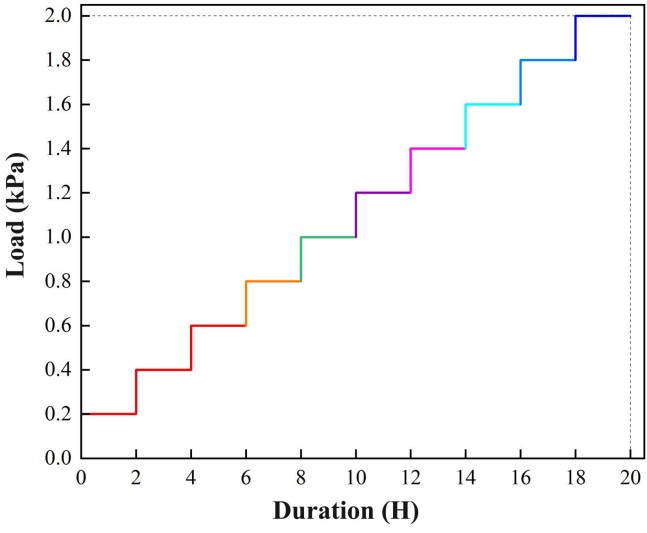

**Fig 14. Loading-time curve for the model tests.**

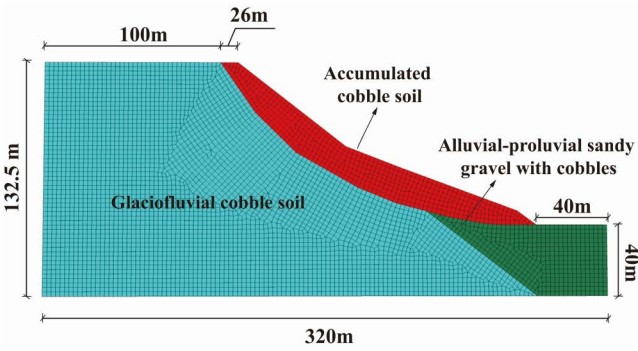

**Fig 15. FLAC³D numerical model of the slope.**

of glaciofluvial pebble soil—were subjected to testing. Both the sliding mass and slip zone soils consist of pebble soil; however, due to the high moisture content of the slip zone layer, considerable sample disturbance occurred during sampling. Therefore, the test results for the slip zone soil were treated as reference data, and the saturated-state parameters were determined in accordance with relevant standards and previous empirical studies. Minor differences were observed among the test results for different slope soil types; consequently, standard values were adopted as the representative parameters. The mechanical parameters of each soil type are listed in Table 8.

In accordance with the specifications defined in the Code for Investigation of Landslide Prevention and Control Engineering (China, GB/T 32864−2016), the shear strength parameters of slip-zone soil can be back-calculated using designated formulas:

$$C = \frac{K_s \sum W_i \sin\alpha_i - \tan\varphi \sum W_i \cos\alpha_i}{L} \tag{7}$$

$$\varphi = \tan^{-1}\left(\frac{K_s \sum W_i \sin\alpha_i - CL}{\sum W_i \cos\alpha_i}\right) \tag{8}$$

In the equation: $K_s$ represents the stability coefficient, with its value varying under different working conditions; $\alpha_i$ denotes the dip angle of the i-th sliding block (°); $c$ refers to the cohesion of the slip-zone rock-soil mass (kPa); $\varphi$ denotes the internal friction angle of the slip-zone rock-soil mass (°); $W_i$ – represents the unit weight of the i-th block (kN/m³).

Based on the above parameter back-calculation formulas and the observed slope deformation in the study area, the slope surface and slip surface were used as the inversion model, and representative cross-sections were selected for back-analysis, as shown in Fig 16. The stability coefficient is 1.05 under natural conditions and 1.00 under heavy rainfall

**Table 8. Key Physical and Mechanical Parameters of Landslide Rock and Soil Materials.**

| Categories of Geotechnical Materials | Density $\rho$ (g/cm³) | | Cohesion c (kPa) | | Internal friction angle $\varphi$ (°) | |
|---|---|---|---|---|---|---|
| | Natural | Saturated | Natural | Saturated | Natural | Saturated |
| Sliding Mass Soil | 2.02 | 2.16 | 9.5 | 6.2 | 31.0 | 28.9 |
| Slip Zone Soil | 1.95 | 2.09 | 9.1 | 5.7 | 27.4 | 25.6 |
| Sliding Bed Soil (Fluvial-Alluvial Deposits) | 2.09 | 2.19 | 9.6 | 6.0 | 31.1 | 28.3 |
| Sliding Bed Soil (Glaciofluvial Deposits) | 2.07 | 2.14 | 10.1 | 6.5 | 33.5 | 30.6 |

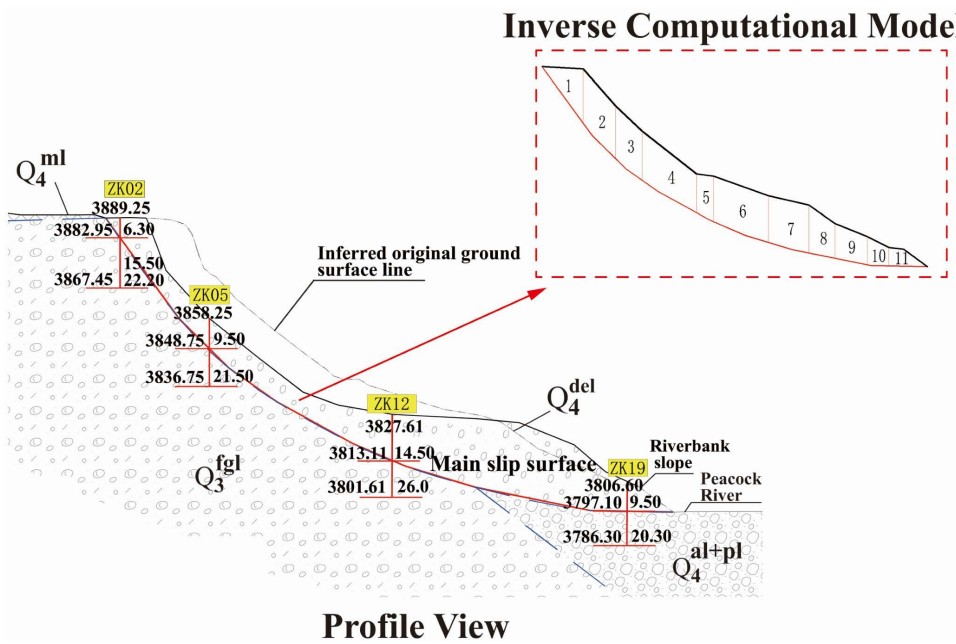

**Fig 16. Figure of the cross-section inversion model.**

conditions. Under these conditions, the shear strength parameters of the slip-zone soil were back-calculated, and the results are summarized in Table 9.

Based on the results of laboratory tests and inversion calculations, the landslide's shear strength test data and inversion-derived values were combined using a 7:3 weighting ratio. With reference to regional empirical data, the integrated physical and mechanical parameters for each part of the landslide were comprehensively determined, as summarized in Table 10.

In summary, the parameters listed in Table 10 were adopted for the numerical simulations in this study. Additionally, to facilitate computation, the model was simplified as follows: because the Quaternary Holocene artificial fill layer ($Q_4^{ml}$) and the Quaternary Holocene cultivated soil layer ($Q_4^{pd}$) display considerable spatial variability, uneven and generally thin thicknesses, their effects were omitted from the numerical model.

For the computational conditions, the numerical model was assumed to consist of a homogeneous soil mass, and the simulations primarily considered the load magnitude and its position at the slope crest. The surcharge area was defined using geometric coordinates or local mesh refinement, after which a uniformly distributed surface load was applied to generate vertical stress, ensuring that the load acted solely on the ground surface without propagating into deeper layers. Based on the surcharge width used in the physical model tests, the loading area was set to a width of 20 m. Six loading combinations were designed, as summarized in Table 11. Since the actual load acting on the in-situ slope could not be obtained, and considering the inherent limitations of the physical model test, the load applied in the numerical simulation was converted from the model test load according to the similarity ratio. The maximum applied load was 400 kPa.

**Table 9. The key physical parameters of the slop.**

| Name of the Slip-Zone Soil | Cohesion $c$ (kPa) | | Internal friction angle $\varphi$ (°) | |
|---|---|---|---|---|
| | Natural | Saturated | Natural | Saturated |
| Cobbly Soil | 6.7 | 4.3 | 23.7 | 22.3 |

**Table 10. Comprehensive table of the key physical parameters.**

| Categories of Geotechnical Materials | Density $\rho$ (g/cm³) | | Cohesion $c$ (kPa) | | Internal friction angle $\varphi$ (°) | |
|---|---|---|---|---|---|---|
| | Natural | Saturated | Natural | Saturated | Natural | Saturated |
| Sliding Mass Soil | 2.02 | 2.16 | 8.7 | 5.7 | 31.0 | 29.7 |
| Slip Zone Soil | 1.95 | 2.09 | 8.4 | 5.3 | 26.3 | 24.6 |
| Sliding Bed Soil (Fluvial-Alluvial Deposits) | 2.09 | 2.19 | 8.8 | 5.5 | 29.8 | 27.2 |
| Sliding Bed Soil (Glaciofluvial Deposits) | 2.07 | 2.14 | 9.3 | 6.0 | 32.1 | 29.6 |

**Table 11. Loading combinations for the slope in the numerical simulations.**

| Distance from slope crest(m) | Applied load(kPa) | | |
|---|---|---|---|
| 10 | 200 | 300 | 400 |
| 20 | 200 | 300 | 400 |

## 3. Result

### 3.1. Experimental results

**3.1.1. Apparent deformation characteristics.** During the initial loading stage (0–10 h), the internal stress within the embankment has not yet reached its shear strength, and the deformation characteristics are shown in Fig 17. At the front edge of the loading area, the relatively steep slope causes substantial soil particle slippage, while localized sliding of soil blocks occurs in the upper slope. Minor settlements and shallow cracks develop on the slope surface, and small tensile cracks appear at the front edge and the upper-middle section. At the rear edge of the loading area, lateral compression induced by the applied load leads to horizontal compression and shear deformation of the soil, resulting in cracks located 10–30 mm from the edge of the loading area.

As time progresses and the applied load increases (10–18 h), slope deformations initiate, with characteristics shown in Fig 18. At the front edge of the loading area, transverse cracks progressively develop, with widths of 5–12 mm, inclinations of 70°-90°, and penetration depths of 10–30 mm. Cracks at the slope crest and face extend inward and laterally, showing marked increases in both width and length. Localized differential settlements and collapses occur on the slope face, especially in steep and fractured soil areas, while cracks begin to appear in the central slope. Microcracks within the slope gradually propagate under concentrated normal stress, giving rise to potential sliding surfaces. At the slope toe, a bulging crack zone forms, with cracks situated 0–20 mm from the lower boundary and 15–20 mm from the lateral edges. This bulging zone extends approximately 800 mm in width, with individual crack widths of 2–5 mm and depths of 5–8 mm, corresponding to the positions of deep-seated shear outlets.

When the loading time reaches 19.1 h (with an applied load of 2 kPa), soil stresses generally exceed the yield strength and reach the failure limit. Cracks continue to develop, propagate, and extend, with central slope cracks progressively enlarging, as shown in Fig 19. Transverse through-cracks at the rear edge of the loading area extend toward the slope toe, where soil bulges and extrudes outward. Under lateral thrust, the slope toe moves forward, and the sliding surface eventually becomes fully continuous, leading to complete slope failure. Tensile cracks at the rear edge penetrate downward, while the shear zone extends into the slope body, together forming a potential sliding surface. At the slope toe, material is sheared off; the front edge forms a tongue-shaped accumulation, while the rear edge develops scarps and steep cliffs due to tension, indicating a total loss of slope bearing capacity.

**3.1.2. Slope stress characteristics.** At 19.1 h, the slope failed, and soil pressure measurements at the monitoring points are shown in Fig 20. The results indicate that vertical stress increments beneath the loading area gradually decrease with depth. Sensors 4 and 3, positioned nearer the loading zone, exhibit pronounced changes, with stress

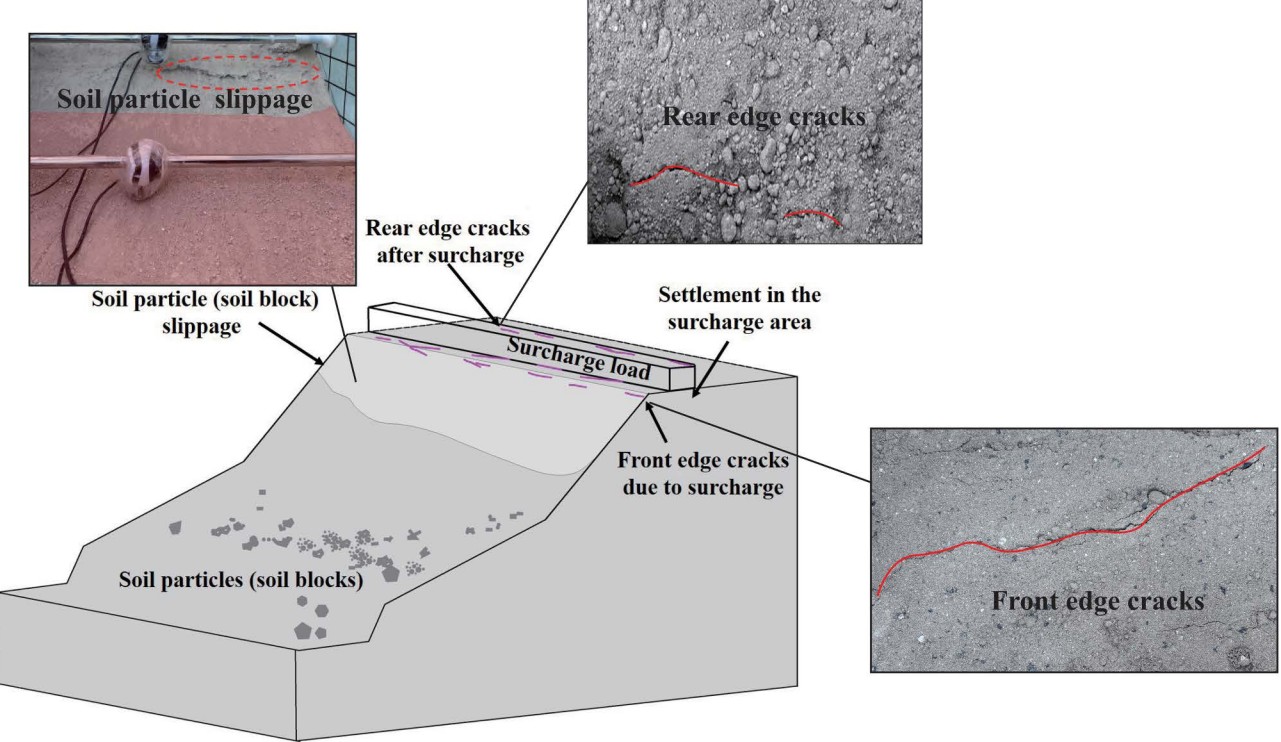

**Fig 17. Slope characteristics during the initial loading stage.**

increments of approximately 0.85 and 0.69 times the applied load, respectively. Sensors 2 and 1, located farther from the loading zone, show smaller variations, with stress increments of about 0.45 and 0.17 times the applied load. At the slope toe, sensor 5 exceeding readings from other sensors at the same depth, with stress increments of approximately 0.92 times the applied load, indicating significant stress concentration at the slope toe. Furthermore, data trends reveal that in the 2–3 hours before failure, all monitoring points display an upward inflection, serving as a clear precursor to slope instability. In summary, under the applied load, the slope's vertical stress distribution exhibits a composite pattern of "gradual attenuation with depth coupled with stress amplification at the slope toe."

**3.1.3. Slope displacement characteristics.** Fig 21 illustrates the evolution of horizontal displacements at the upper, middle, and lower sections of the slope during the loading process. The results indicate that displacements at sensors 3 and 2 gradually increase over time. During the first 14 h, displacement growth is slow; after 14 h, the rate of displacement increases markedly, entering a rapid deformation stage. At 17 h, the displacement at sensor 2 reaches its peak. Throughout the loading period, displacement at sensor 2 is consistently greater than at sensor 3, indicating that deformation in the middle slope is significantly larger than at the slope toe. This suggests that the steeper central region of the slope is more sensitive to the applied load, with more pronounced deformation development.

Notably, the area around sensor 1 experienced collapse during the test, resulting in missing displacement data after 12 h, which prevents effective comparison. Comparison of the initial and final coordinates of the monitoring points indicates that vertical displacements along the slope height decrease progressively, with the maximum vertical settlement occurring at the crest of the loading zone in the middle slope.

The variations in data from nine monitoring points located in three typical regions—the front edge of the loading area (near the free face), the loading area, and the region between the front edge and the loading area—are shown in Fig 22

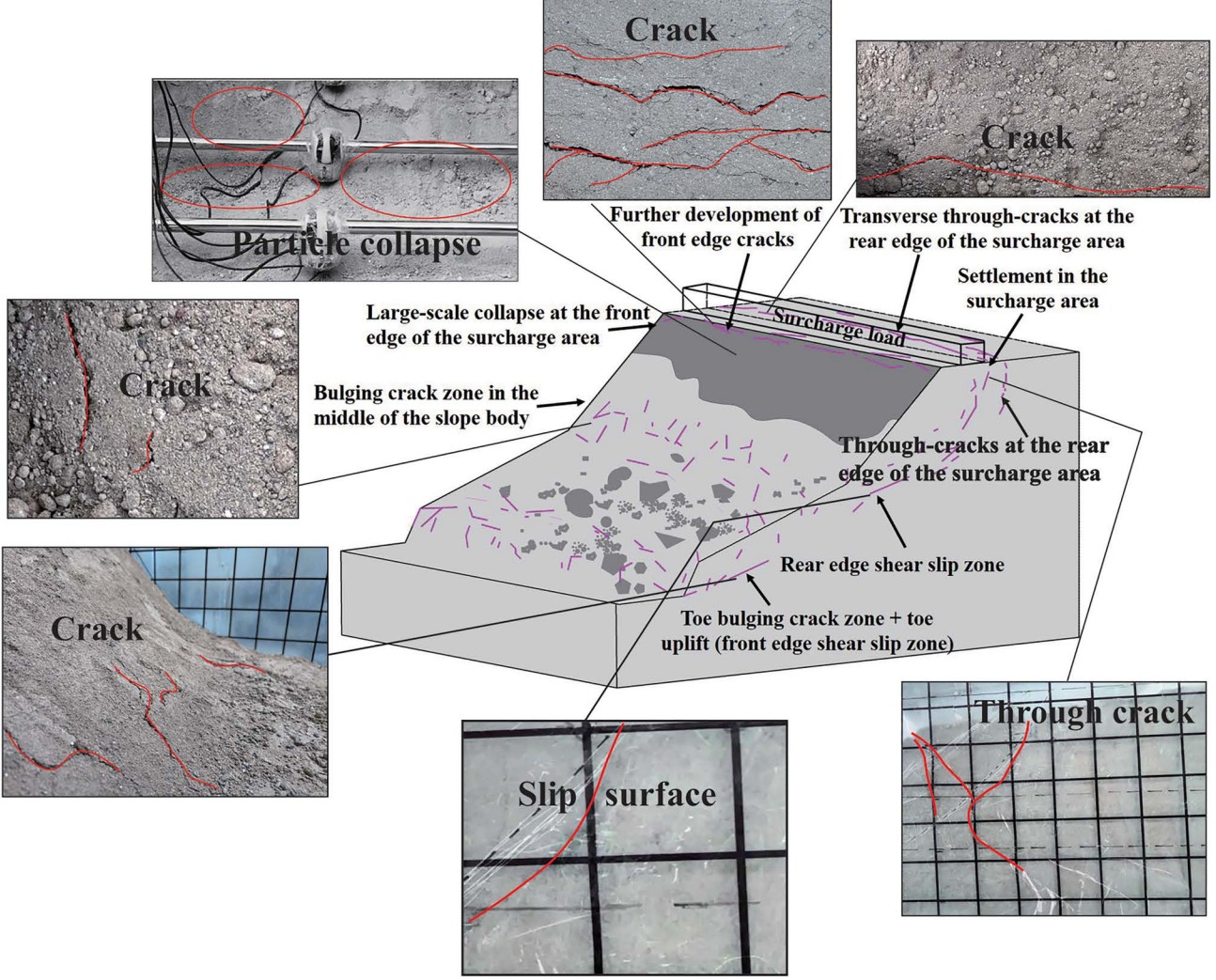

**Fig 18. Slope characteristics during the deformation development stage.**

(see Fig 12 for locations). The results indicate that vertical displacements in the loading area and the region between the front edge and the loading area gradually decrease from the slope crest to the slope toe, whereas horizontal displacements at the front edge of the loading area initially increase from the crest downward and then decrease toward the slope toe. According to the vertical coordinates of the installed monitoring points, vertical displacements in the loading area, the front edge area, and the intermediate region between them decrease progressively from the slope crest to the slope toe.

Displacement at the slope crest primarily manifests as vertical settlement, with the maximum settlement occurring at the crest of the loading area in the middle slope, based on the displacement variation data from the typical regional monitoring points, the maximum vertical displacement was 40 mm at monitoring point C3, and gradually decreasing with depth.

Displacements along the upper, middle, and lower slope face mainly occur in the horizontal direction, with bulging observed in the middle and lower slope sections; the steep central region exhibits the largest horizontal displacement. According to extensometer data, the maximum horizontal displacement in the upper slope remains unclear during the loading process, whereas the maximum horizontal displacement in the middle slope is 26 mm, and at the slope toe it is

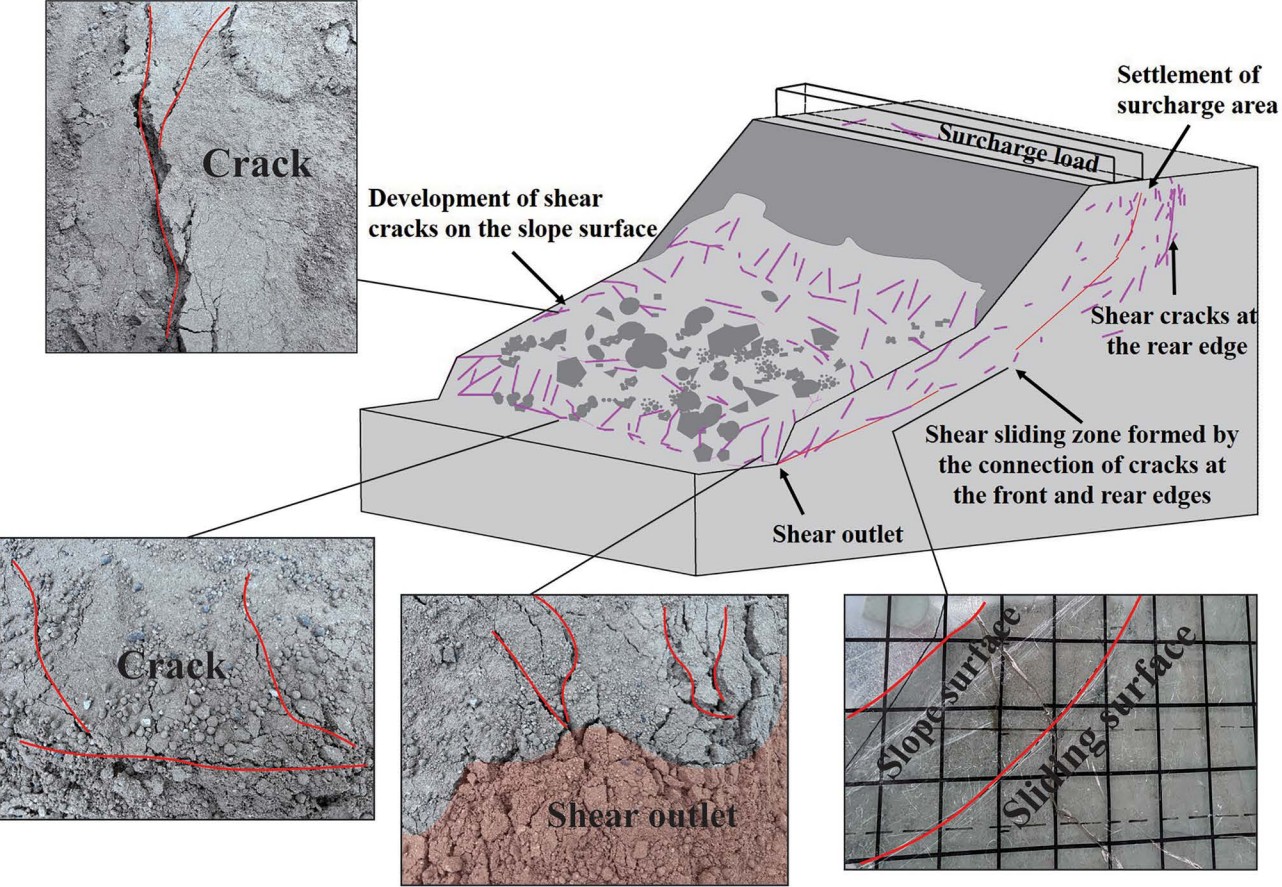

**Fig 19. Slope characteristics during the deformation development stage.**

12 mm. Based on the displacement variation data from the typical regional monitoring points, the maximum horizontal displacement at point A3 in the upper part of the slip zone was 7 mm, while point A2 in the middle part recorded a maximum horizontal displacement of 50 mm, and point A1 in the lower part exhibited a maximum horizontal displacement of 18 mm.

## 3.2. Numerical simulation results

### 3.2.1. Increment of shear strain.
Regions exhibiting peak shear strain increments represent the weakest zones within the slope. The spatial location, geometry, thickness variation, and developmental trends of these shear strain increment zones can, to a certain extent, directly reflect the characteristics of sliding zone development within the landslide mass. The shear strain increment cloud under loading conditions is shown in Fig 23.

The results indicate a clear positive correlation between the maximum shear strain increment and the applied load, with concentrated regions located near the slope crest line and beneath the loading area, extending toward the slope toe. For the same applied load, the closer the loading position is to the slope crest line, the more easily a high shear strain increment zone forms. This indicates that the loading location affects the efficiency of load transmission; the shorter the distance, the more pronounced the internal stress concentration, and the faster the potential sliding surface propagates and develops.

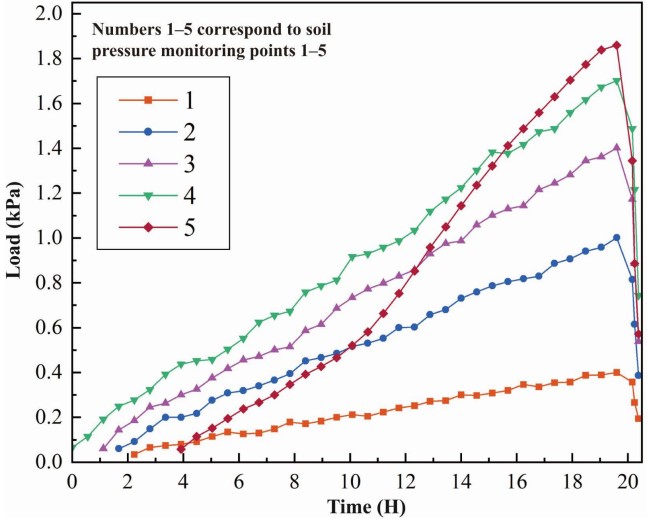

**Fig 20. Variation of soil pressure at monitoring points 1-5.**

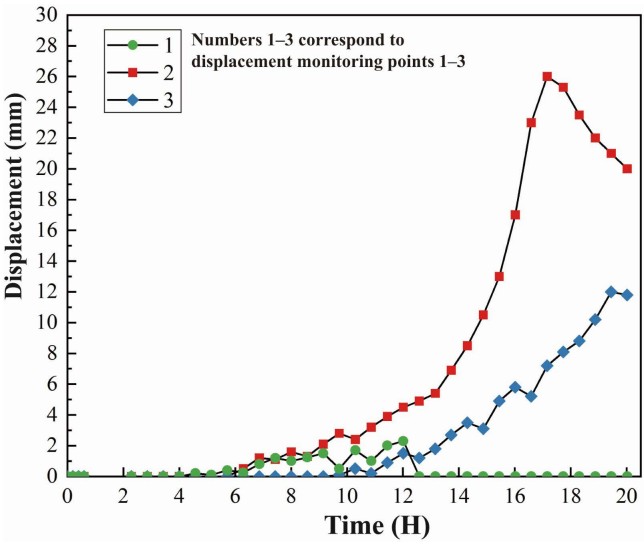

**Fig 21. Displacement evolution recorded by sensors 1-3.**

When the loading position is the same, increasing the applied load leads to a continuous rise in shear strain increments. The high shear strain increment zones gradually extend toward the central and deep regions of the slope, the slope toe, and the periphery of the loading area, resulting in more unstable regions and an expanded influence range on the slope's shear deformation.

**3.2.2. Horizontal displacement.** Displacement is a key indicator for evaluating landslide movement characteristics, as illustrated by the displacement cloud in Fig 24. The results indicate that under stress induced by the applied load, slopes with favorable free-face conditions (steeper terrain in the middle slope) are more prone to displacement, with relatively large horizontal displacements. Comparison with the shear strain increment cloud reveals that the horizontal

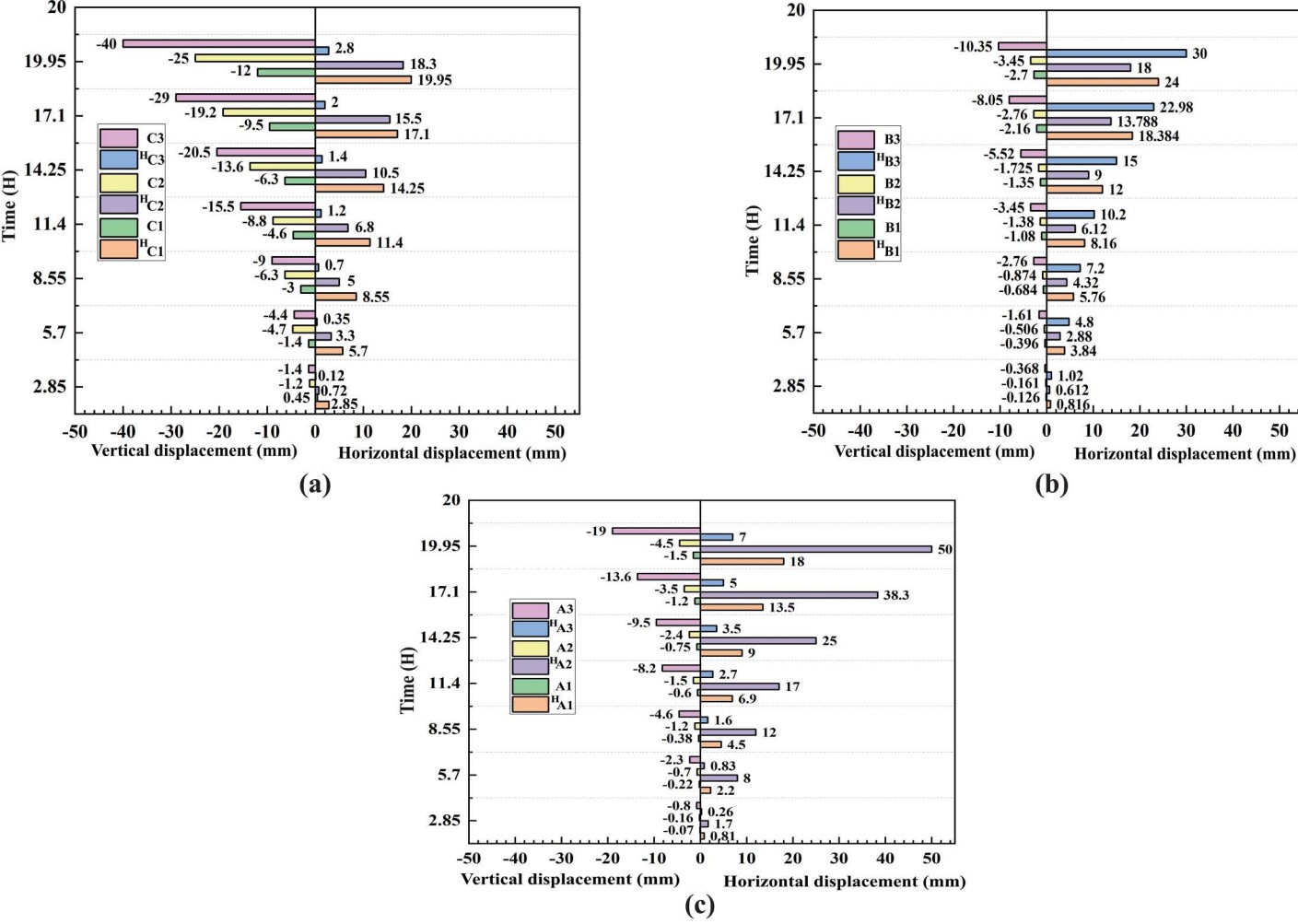

**Fig 22. Displacement evolution at monitoring points in three representative regions: (a) displacement at loading area points; (b) displacement at rear-edge points of the loading area; (c) displacement at front-edge points of the loading area. The superscript "H" denotes the horizontal displacement at the corresponding point.**

displacement directions on either side of the influence zone are opposite. Furthermore, both the position and magnitude of the applied load significantly influence the slope's horizontal displacement. Under the same load, the closer the loading position is to the slope crest line, the more pronounced the stress concentration; due to weaker lateral constraints, the soil more readily moves toward the free face, increasing horizontal displacement. At the same location, higher loads produce higher stresses, and once the elastic limit is exceeded, plastic deformation intensifies, further increasing horizontal displacement.

**3.2.3. Plastic zone analysis.** In the numerical analysis, a detailed examination of the plastic zones enables effective identification of the deformation characteristics of the slope body and provides insight into the stress state of individual slope elements. This analysis allows a clear distinction between tensile and shear failure regions within the slope. Fig 25 presents the distribution of plastic zones under different working conditions.

According to the figure, tensile plastic zones appear at both the front and rear edges of the surcharge area at the slope crest, indicating that the soil in this region primarily undergoes tensile failure. Under the same surcharge,

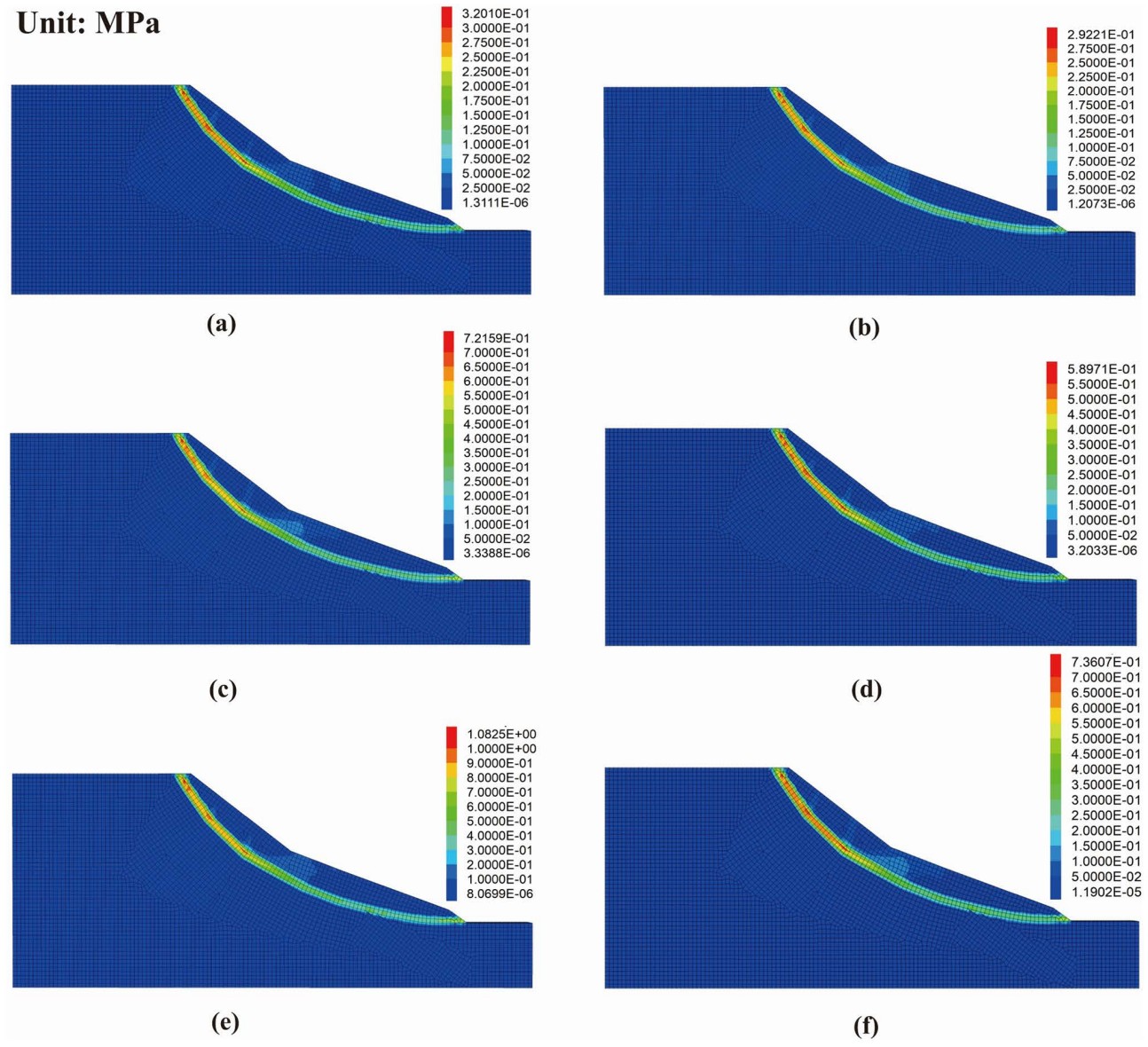

**Fig 23. Shear strain increment contours of the slope model under different conditions: (a) distance 10 m, load 200 kPa; (b) distance 20 m, load 200 kPa; (c) distance 10 m, load 300 kPa; (d) distance 20 m, load 300 kPa; (e) distance 10 m, load 400 kPa; (f) distance 20 m, load 400 kPa.**

shear plastic zones develop within the surcharge area, in the underlying slope body, and at the slope toe. This phenomenon indicates that the soil in these regions fails predominantly through shear deformation, meaning that the soil beneath the surcharge area and at the slope toe gradually experiences failure due to shear forces induced by the surcharge.

Furthermore, due to the stress transfer and redistribution induced by the surcharge, the shear stress acting on the soil near the surcharge area gradually increases. Once this stress exceeds the soil's shear strength, a shear failure zone forms and progressively extends toward the middle of the slope as the surcharge persists or related influencing factors continue. Under the combined effects of the surcharge and the self-weight of the

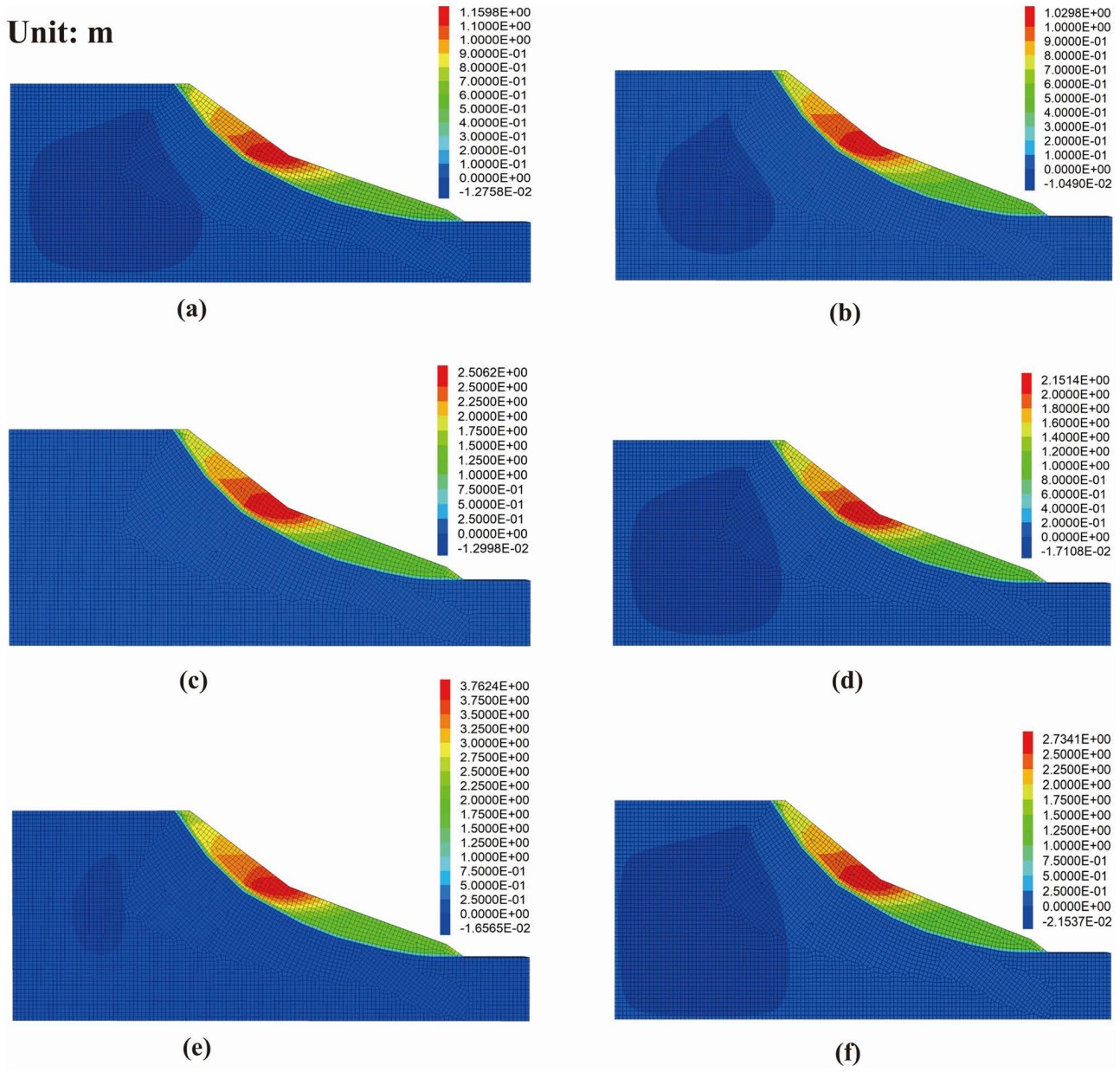

**Fig 24. Horizontal displacement contours of the slope model under different conditions:** (a) distance 10 m, load 200 kPa; (b) distance 20 m, load 200 kPa; (c) distance 10 m, load 300 kPa; (d) distance 20 m, load 300 kPa; (e) distance 10 m, load 400 kPa; (f) distance 20 m, load 400 kPa.

slope, the slope toe is subjected to elevated shear stress. When this stress surpasses the soil's shear strength, shear failure initiates at the slope toe and subsequently propagates upslope toward the slope's central region. Consequently, shear failure zones originating beneath the surcharge area and at the slope toe expand progressively, resembling two extending "failure bands" that advance toward the middle of the slope. Over time, these bands eventually intersect and merge, further intensifying soil failure and posing a significant threat to the overall stability of the slope.

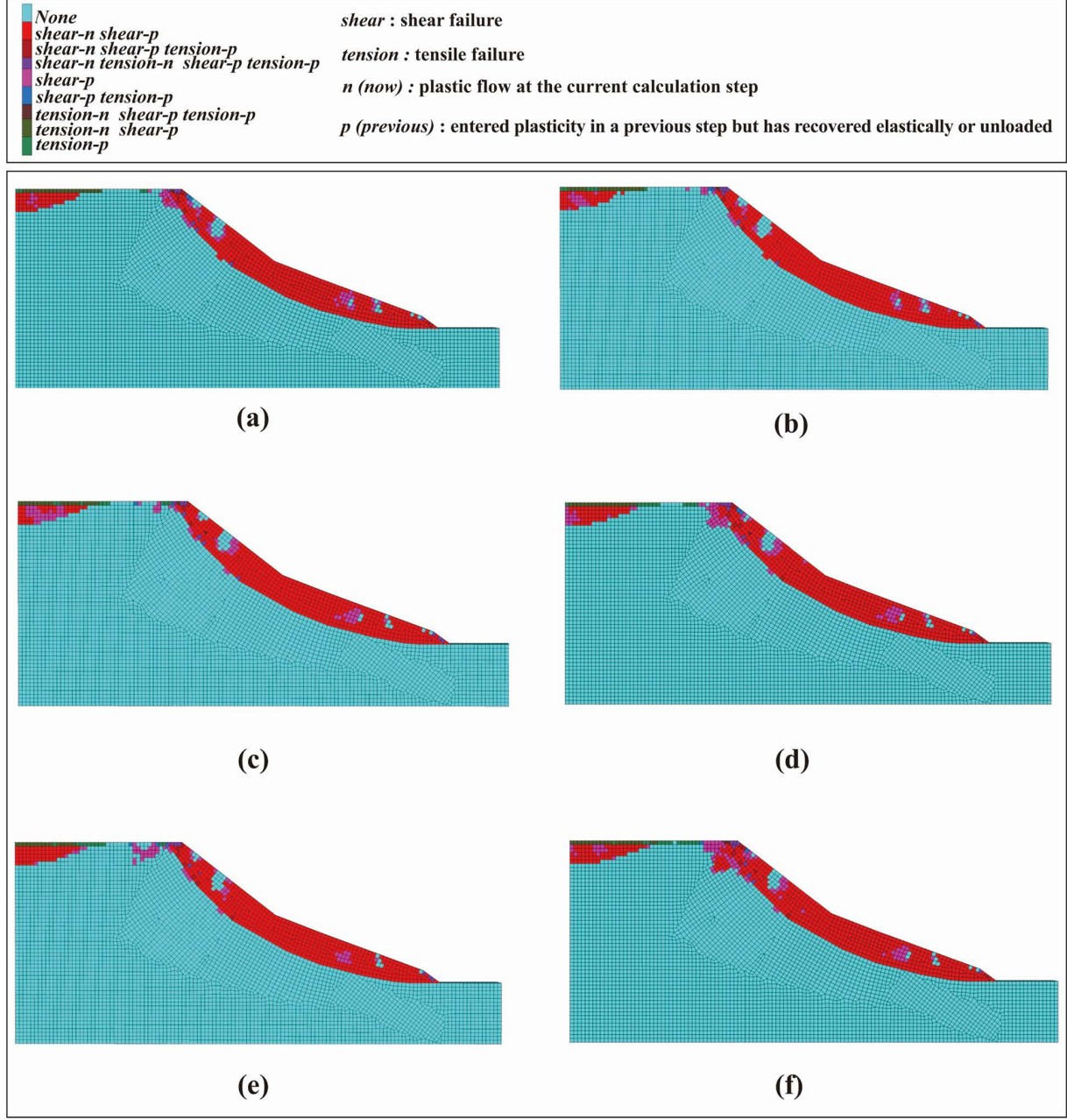

**Fig 25. Plastic Zone under different conditions:** (a) distance 10 m, load 200 kPa; (b) distance 20 m, load 200 kPa; (c) distance 10 m, load 300 kPa; (d) distance 20 m, load 300 kPa; (e) distance 10 m, load 400 kPa; (f) distance 20 m, load 400 kPa.

## 4. Discussion of results and summary of deformation and failure mechanisms

### 4.1. Discussion of results

Both experimental and numerical simulation results indicate that slope deformation under applied loading gradually propagates from the top toward the middle section. As the applied load increases, the high shear strain increment zones continuously migrate toward the slope middle, revealing a pronounced positive correlation between load magnitude and

the spatial extent of shear deformation. The underlying mechanism is that the shear stress induced by the loading continuously intensifies, causing soil initially in an elastic or slightly damaged state to progressively enter the shear failure stage, thereby driving the evolution of the failure zone toward the slope middle and ultimately facilitating progressive instability.

Experimental results show that soil deformation is most pronounced near the loading area, a feature further confirmed by numerical simulations. Under the same load, the extent of shear failure decreases with increasing distance from the slope crest line, indicating that stress concentration is particularly pronounced near the crest, where soil more readily enters the shear failure stage and the failure zone forms first. Meanwhile, the slope toe experiences higher shear stress under the combined action of loading and self-weight. When this stress exceeds the soil's shear strength, failure occurs and gradually propagates toward the slope middle. Consequently, the shear failure zones typically migrate simultaneously from beneath the loading area and from both ends of the slope toe toward the slope middle, eventually connecting over time to form a continuous failure band. This indicates that the slope crest line and the slope toe are the most sensitive regions for slope stability in the embankment.

In addition, the middle section of the slope with favorable free-face conditions exhibits more pronounced horizontal displacement. This is primarily attributed to insufficient lateral restraint under loading and the steeper terrain, which induces stress concentration and facilitates soil movement toward the free face. Simultaneously, the horizontal displacement directions of soil inside and outside the shear strain increment zone are opposite. This is due to significant differences in stress states: soil within the zone undergoes stronger compression and shear, resulting in horizontal displacements opposite to those of the soil outside the zone under combined gravitational effects.

## 4.2. Embankment Slope Deformation and Failure under Surcharge Loading

Based on physical model experiments and numerical analysis, this study summarizes the failure mechanism of deposit slopes under applied loading, which can be divided into three main stages, as shown in Fig 26.

During the initial loading stage, the slope undergoes stress redistribution, with internal stresses in the soil not yet reaching shear strength. This stage is characterized by localized minor settlements and shallow cracks on the slope surface, with some soil blocks slipping. Local tensile cracks appear at the crest and front edge of the loading area, while soil at the rear edge experiences lateral compression, resulting in horizontal compression and shear deformation.

As time progressed and the surcharge load continued to increase, the maximum earth pressure at Monitoring Point No. 4 near the slope crest reached 1.88 kPa, while that at Monitoring Point No. 5 near the slope toe reached 1.85 kPa, exceeding the soil's yield strength and triggering slope deformation and failure. The failure was mainly characterized by progressive crack propagation and penetration, accompanied by the formation of an internal slip surface. Uplift occurred at the slope toe within the surcharge zone, and Displacement Gauge No. 3 recorded a maximum horizontal displacement of 12 mm, indicating substantial lateral thrust that induced shear failure and produced a tongue-shaped accumulation body. According to the displacement data, the maximum horizontal displacements at Points A3 (upper part of the slip belt), A2 (middle part), and A1 (lower part) were 7 mm, 50 mm, and 18 mm, respectively. A scarp and steep face developed at the rear of the surcharge zone, signifying a complete loss of the slope's bearing capacity. Overall, the failure mechanism involved stress concentration that initiated and propagated cracks, the formation of a potential slip surface, and ultimately toe bulging, shearing, and overall sliding failure. In summary, the formation and evolution of slope failure under applied loading follows a "progressive migration" mechanism: loading→initial equilibrium disruption→rear-edge tensile cracking→upper soil sliding→front-edge compression and bulging→sliding surface propagation and interconnection→overall slope sliding failure.

## 5. Comparative analysis

### 5.1. Studies for comparison with the present work

To validate the rationality of the deformation and failure mechanism proposed in this study and to highlight the distinct deformation and failure characteristics of accumulation slopes, Table 12 compiles previous research on slope deformation and failure under additional loading for comparative analysis.

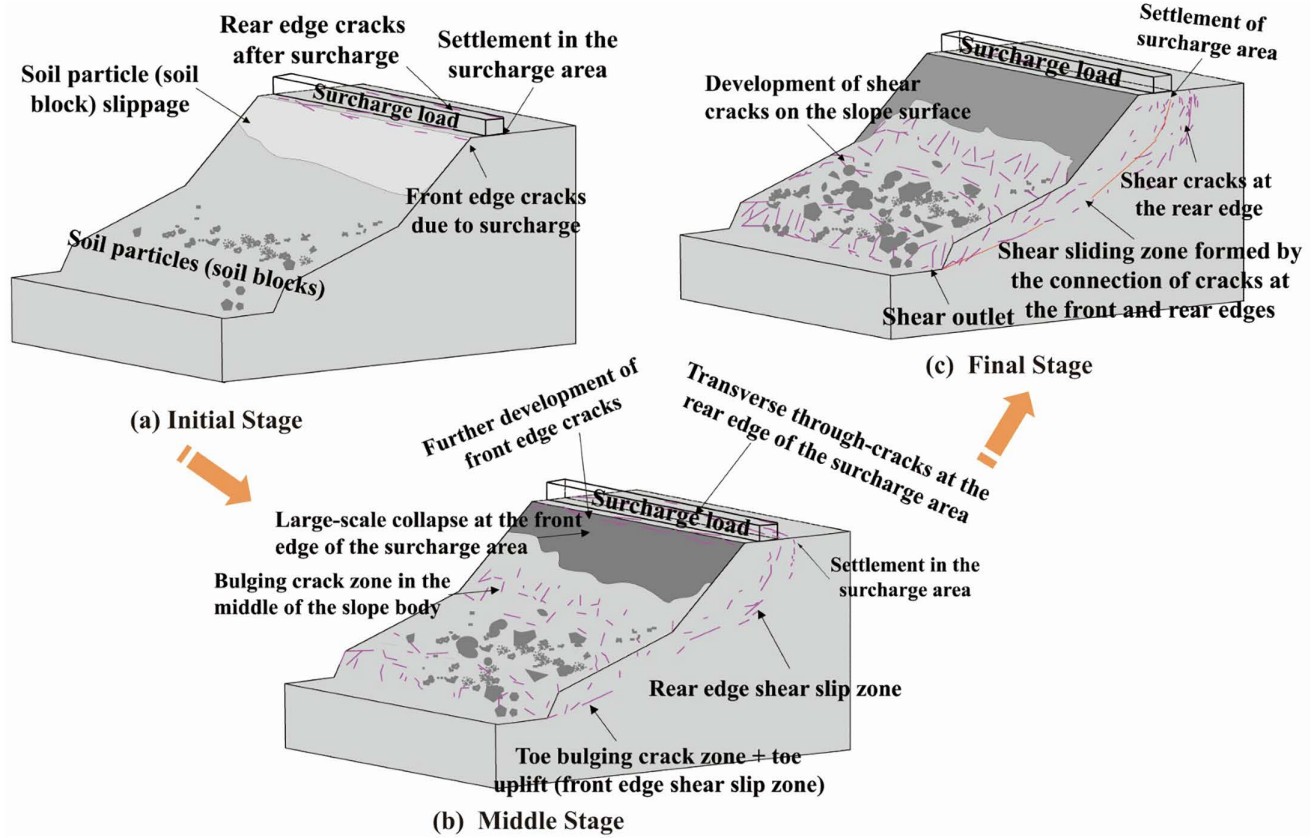

**Fig 26. Three stages of deformation and failure in an accumulation slope.**

**Table 12. Comparison of slope deformation and failure modes under external loading.**

| Source | Category | Load type | Loading position | Research method |
|--------|----------|-----------|------------------|-----------------|
| [56] | Homogeneous slope | Uniform load, trapezoidally distributed load and concentrated load | Slope crest | Numerical analysis |
| [57] | Homogeneous slope | Uniform load, embedded load, concentrated load | Slope crest | Numerical analysis combined with experimental testing |

Ref. [56] examined the deformation and failure behavior of slopes subjected to additional loading using numerical analysis. The deformation-failure pattern under a uniform load is illustrated in Fig 27a. The findings show that, once a uniform load is applied to the slope crest, the slope initially enters an elastic response stage, during which overall deformation remains small and no clear indicators of failure emerge. With continued increases in load magnitude, a shear strain concentration zone develops in the shallow region near the slope crest and gradually evolves into a localized shear band, marking the transition from the elastic stage to the elasto-plastic stage. During this stage, the shallow shear band extends in a strip-like form along the slope surface, representing a characteristic feature of the initial failure process.

As the external load continues to increase, the shallow shear band progressively extends downward and eventually connects to form a continuous weak shear surface, accompanied by a marked enlargement of the plastic zone within the slope. The failure zone follows a characteristic shallow-to-deep evolutionary pattern: it initiates with localized damage in

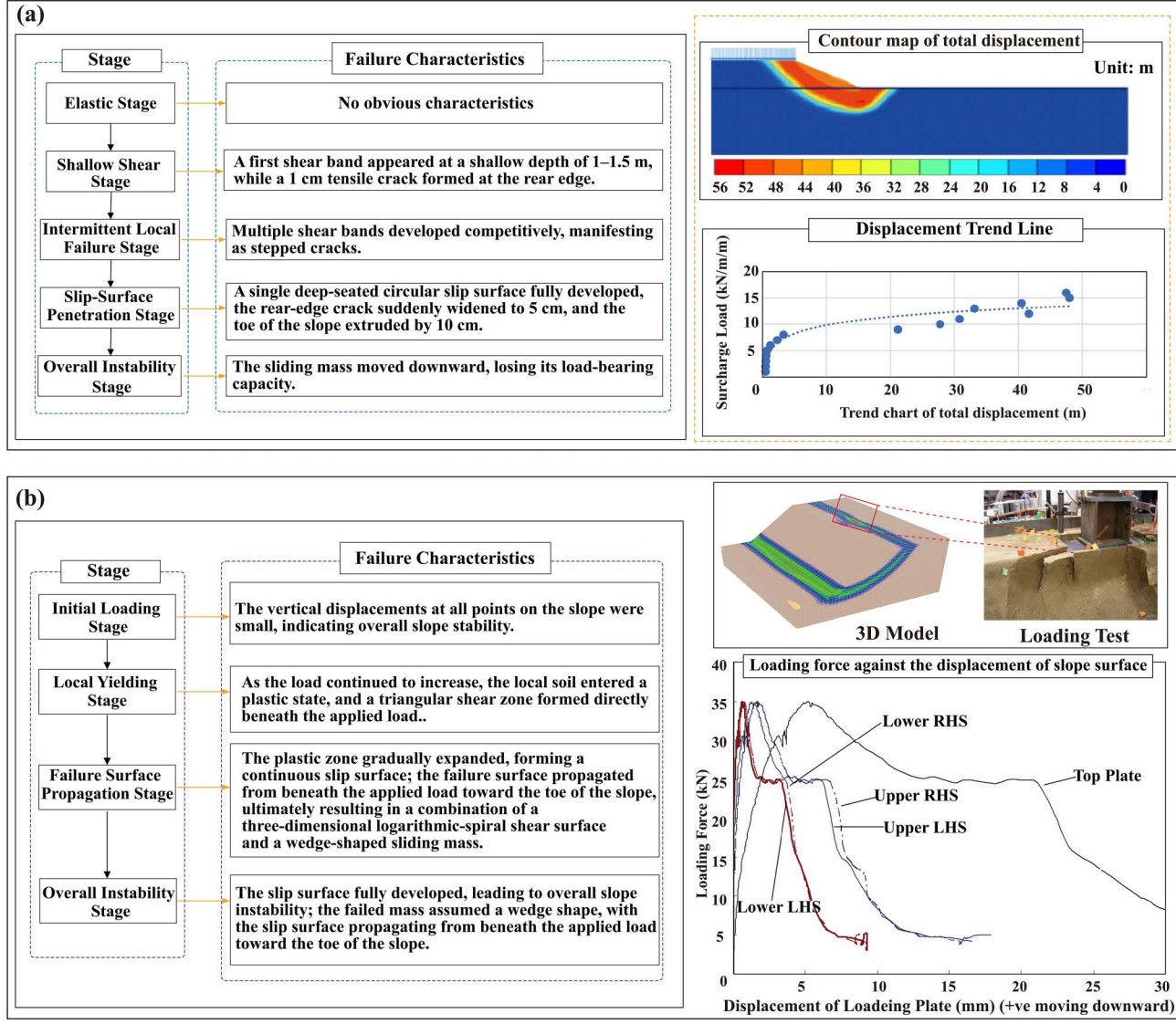

**Fig 27. Failure modes of a homogeneous slope subjected to uniform loading: (a) Failure mode of the slope under a uniform load; (b) Failure mode of the slope under a locally applied uniform load.**

the shallow portion of the slope surface, advances gradually into the deeper interior, and ultimately develops into a deep-seated sliding surface. This progression is also reflected in the displacement-load curve, which exhibits a rapid increase in slope as the load approaches its ultimate value, indicating a significant reduction in slope stiffness and the onset of accelerated instability. Displacement contours from the numerical simulation further demonstrate an overall sliding deformation extending from the slope crest to the toe, with deep-seated mass movement constituting the dominant failure mode. Failure induced by a uniform load is characterized by a large affected zone, parallel shear-band development, and a progressively deepening slip surface, highlighting the broad and integral nature of slope failure under uniform loading.

Ref. [57] examined the three-dimensional slope stability and associated failure mechanisms when a locally applied uniform load was imposed at the slope crest. The deformation and failure pattern under the local uniform load is shown in Fig 27b. The findings indicate that, unlike the widespread shallow failures induced by a fully uniform load, the slope failure

triggered by a local load exhibits pronounced localization and directionality. During the initial loading stage, a distinct stress concentration and the onset of a plastic zone first occur beneath the loaded area, manifested in the rapid development of a localized shear band. The peak shear strain within this zone is substantially higher than that in the surrounding soil, causing the local structural integrity of the slope to be compromised first.

As the load continues to act and the plastic zone expands, the shear band progresses into the slope interior and downward in a wedge-shaped or tongue-shaped pattern, exhibiting pronounced three-dimensional propagation characteristics. Because the loaded area is limited, the failure no longer develops uniformly along the slope surface but instead evolves into an asymmetric shear-failure path aligned with the direction of stress concentration. The local load also tends to induce tensile cracking at the slope crest. These cracks widen as deformation increases and gradually propagate toward the slope surface, with their trajectory consistent with the contour of the internal shear band, further confirming the localized and directional nature of the failure.

When the localized shear zone expands to a critical scale, the plastic zone extends toward the slope toe and becomes fully connected, forming a composite shear-sliding surface and marking the onset of overall slope instability. At this stage, the displacement curves show a rapid increase in the displacement difference between the upper and lower segments of the sliding surface, indicating that the sliding mass is transitioning into an accelerated downslope movement phase.

Compared with a uniformly distributed load, failure induced by a local load is more strongly governed by stress-concentration points. Although the affected zone is relatively limited, the resulting damage is more intense, making rapid localized failure more likely and allowing it to propagate progressively to the entire slope. Overall, under a locally applied uniform load, slope failure exhibits features such as point-to-surface evolution, pronounced spatial directionality, and three-dimensional wedge-shaped expansion of the shear band, demonstrating the strong perturbation effect of local loading on slope stability.

### 5.2. Comparative analysis results

#### 5.2.1. Comparison of failure mechanisms in homogeneous slopes under additional loading. This study compares the slope failure patterns identified herein with those reported in previous studies under uniformly distributed loads and locally applied uniformly distributed loads. The results show that slopes composed of colluvial or accumulative deposits exhibit a characteristic "pushing-type" failure mode when subjected to surcharge-induced disturbances. These failure features arise not only from the inherent properties of the deposits—namely weak cementation and low shear strength—but also from the pronounced influence of load type, stress-transfer pathways, and modes of spatial propagation.

At the initial stage of failure evolution, the slope was primarily undergoing stress redistribution, characterized by the emergence of shallow short cracks, the formation of tensile cracks at the rear edge of the surcharge, and localized settlement near the slope toe. The characteristics of this stage align with the early deformation patterns reported under uniformly distributed loading in Reference 57, which similarly show that shallow stress perturbations develop first. However, unlike the planar and uniformly applied disturbance produced by a uniformly distributed load, the surcharge in this study functions as an approximately "locally concentrated load". Consequently, the rear-edge tensile cracks are more pronounced and display distinct structural-control features, more closely matching the initiation behavior of localized shear zones induced by locally applied uniformly distributed loads as described in Ref. [57].

As the surcharge continues to act, the slope transitions into a stage of intensified deformation. The number and width of cracks increase markedly, and the crack network expands both inward and laterally within the slope. Penetrating microcracks develop within the interior and gradually evolve into a potential slip surface. This process is consistent with the "shallow-intermediate-deep progressive extension" failure pathway reported under uniformly distributed loading, reflecting the gradual formation of the slip surface. However, the colluvial slope simultaneously exhibits a pronounced coupled response characterized by frontal bulging and rear-edge downward displacement. The relative movement of the sliding

mass becomes more evident, representing a typical pushing-type structural failure. In contrast, shear bands formed under uniformly distributed loads tend to be more parallel and widespread, whereas those generated by localized loading display stronger directionality and wedge-shaped propagation. Therefore, the failure mode observed in this study falls between the global extension produced by uniformly distributed loads and the directional extension induced by localized loads, but it more closely resembles failure patterns triggered by localized loading.

At the final failure stage, all three types of slopes exhibit a fully developed slip surface and overall instability of the sliding mass. Under uniformly distributed loading, the slip surface penetrates deeply and continuously, producing large-scale downslope displacement. Under localized loading, the shear surface typically shows three-dimensional wedge-shaped or tongue-shaped propagation. The pushing-type failure identified in this study is characterized by a distinct rear scarp, a bulging shear-out zone at the slope front, and a through-going slip surface that develops from top to bottom, ultimately forming an integrated sliding accumulation mass. Although this failure mode aligns more closely with the directional failure induced by localized loading, its progressive evolution is more complex, exhibiting a multi-stage coupled sequence: initiation of rear-edge tensile cracking→sliding of the upper-middle block→frontal shear-bulging extrusion→penetration of the slip surface→overall sliding.

Overall, the failure mechanism of colluvial slopes is jointly governed by the structural characteristics of the colluvial material, the looseness of joints, and the strong stress-concentration effects induced by localized surcharge loading. In terms of failure evolution, the slopes exhibit both the "shallow-to-deep" shear-extension pattern typical of uniformly distributed loading and the "point-to-surface" wedge-shaped directional propagation associated with localized loading. However, the final evolution of colluvial landslides tends to develop into a typical pushing-type failure, in which the failure process is more strongly influenced by the surcharge location, surcharge width, and the stratification of the colluvial deposits.

Therefore, the failure mechanism of colluvial slopes under surcharge loading cannot be simply categorized as either the uniformly distributed loading-induced mode or the localized loading-induced mode. Rather, it should be understood as a surcharge-induced cumulative, pushing-type, and structurally controlled failure mechanism that develops on the basis of the inherent structural weakness of the colluvial deposits.

**5.2.2. Comparison of failure mechanisms in colluvial slopes under rainfall and seismic loading.** For the rainfall-induced failure of colluvial deposits, the comparative reference used in this study is Reference 16. Khan et al. [16] investigated the failure mechanism of unsaturated coal-gangue colluvial slopes subjected to rainfall infiltration. The failure process induced by rainfall infiltration in coal-gangue slopes evolves through two distinct stages. The first stage is "seepage-driven saturation and suction loss", during which rainwater rapidly infiltrates through the highly permeable gangue material, causing matric suction to decrease sharply as the water content increases. Due to its highest permeability, the slope crest becomes saturated first, leading to nonuniform swelling and the formation of tensile cracks. The second stage is "strength degradation and initiation of sliding". As seepage continues, pore-water pressure shifts from negative to positive, effective stress progressively decreases, and the shear strength along the potential slip zone significantly deteriorates. Under the combined effects of self-weight and seepage forces, a traction-type sliding initiates from the crest and propagates downslope, ultimately reducing the factor of safety to below 0.9 and resulting in global slope failure.

For earthquake-induced failures in colluvial deposits, Ref. [23] is used as the comparative source in this study. Ma et al. examined the failure mechanisms of colluvial slopes through shaking-table experiments. The seismic failure process of colluvial-layer landslides can be summarized in two stages. The first stage is "crack initiation and propagation". As seismic waves travel upward from the base of the slip zone, the combined effects of slip-zone strength softening and topographic amplification lead to the formation of tensile cracks in the mid-slope region. These cracks then propagate toward the slope toe, outlining the initial geometry of a throughgoing slip surface. The second stage is "slip-surface formation and instability onset". With continued shaking, the soil within the slip zone undergoes plastic yielding, the point safety factor decreases overall and exhibits a delayed peak, and a deep tensile crack forms at the rear of the sliding mass. The sliding mass

subsequently undergoes a pushing-type displacement from the mid-slope toward the slope toe, ultimately resulting in global failure as the shear resistance along the slip zone drops sharply.

The failure mechanisms of colluvial slopes under different external disturbances exhibit both distinct differences and shared characteristics. Under rainfall conditions, landslide failure progresses through two stages: rainwater first infiltrates rapidly through highly permeable zones, causing early saturation at the slope crest and the development of tensile cracks; subsequently, rising pore-water pressure leads to strength degradation along the slip zone, inducing traction-type sliding from the crest toward the toe. Under seismic loading, failure initiates with tensile cracking in the mid-slope region, after which the slip zone undergoes plastic yielding, and the sliding mass experiences a pushing-type displacement from the mid-slope toward the slope front. In contrast, slope failure induced by surcharge loading in this study exhibits a typical "pushing-type" evolution. Surcharge-driven stress redistribution leads sequentially to rear-edge tensile cracking, crest sliding, and frontal bulging, during which the potential slip surface progressively develops and ultimately forms a tongue-shaped sliding mass.

From a mechanistic perspective, rainfall-induced landslides are primarily controlled by increases in pore-water pressure and the degradation of shear strength, whereas earthquake-induced landslides are driven by instantaneous shearing and plastic yielding. In contrast, surcharge-induced landslides are triggered by stress accumulation and localized stress concentration, exhibiting progressive failure with pronounced structural-control features. The failure process under surcharge loading involves both the "shallow-to-deep" shear propagation characteristic of uniformly distributed loads and the wedge-shaped "point-to-surface" propagation induced by localized loads, while more prominently displaying a multistage interaction consisting of rear-edge tensile cracking, translational sliding of the upper slope, and frontal bulging. Therefore, the failure mechanism of colluvial slopes subjected to surcharge loading is unique: it differs from the traction-type sliding induced by rainfall and from the pushing-type instability caused by earthquakes, instead manifesting as a cumulative-pushing-structurally controlled composite failure mode.

**5.2.3. Failure mechanisms in comparison with rock slopes.** In addition, colluvial slopes differ markedly from rock slopes, with their failure patterns more strongly governed by material weakness and structural looseness. Because rock masses possess higher structural integrity and more regular joint-fracture networks, the failure of rock slopes is typically controlled by geometrically well-defined combinations of structural planes, resulting in structural instability modes such as block sliding, wedge failure, or toppling, with the failure surface being relatively predictable [58]. In contrast, colluvial slopes consist of rock fragments, soil materials, and loose accumulations, and their internal structure is highly heterogeneous. Their deformation generally develops through progressive plastic propagation, and their sliding surfaces tend to be more tortuous, less continuous, and more susceptible to local disturbances. Compared with the "structure-controlled overall sliding" commonly observed in rock-slope failures, colluvial slopes more readily initiate microcracks at sites of local stress concentration and progressively evolve along weak, poorly persistent zones under sustained surcharge loading, ultimately forming a pushing-type overall slide.

Meanwhile, the differing responses of rock slopes under uniformly distributed loading and local loading are mainly reflected in the extent to which structural planes participate in deformation and in the geometric variations of shear bands. In contrast, because particle contacts within colluvial slopes are inherently loose, local surcharge loading more readily induces pronounced stress-concentration effects, resulting in coupled deformation characterized by rear-edge tension cracking, front-edge bulging, and segmented displacement of the sliding mass. This leads to an overall evolutionary pattern of "zonal response-coupled failure" [58].

In summary, from an engineering perspective, colluvial slopes show pronounced failure sensitivity under surcharge loading. The progressive development of rear-edge tension cracking, front-edge bulging, and deep shear bands should be regarded as key monitoring indicators. Controlling surcharge intensity, limiting the spatial extent of surcharge loading, mitigating stress concentration at the surcharge-slope interface, and improving the shear strength of both shallow and deep layers of the colluvial deposits are essential measures for preventing surcharge-induced landslides. In landslide-prone regions, dynamic monitoring systems can be employed for continuous observation [59].

## 6. Conclusion

This study aims to systematically elucidate the deformation-response characteristics and failure mechanisms of colluvial slopes subjected to surcharge loading. By integrating physical model tests with FLAC3D numerical simulations, the study provides a comprehensive investigation of surcharge load transfer, stress distribution, deformation paths, and failure-process evolution. Furthermore, a multivariate regression-based similarity-material proportioning method is proposed, offering systematic theoretical and methodological support for stability analyses of colluvial slopes under artificial loading. The main conclusions are as follows:

(1) Under surcharge loading, the maximum vertical displacement of 40 mm occurs at the slope crest, while the maximum horizontal displacement of 50 mm develops in the steeper mid-slope region. The influence depth of vertical stress is substantially greater than that of horizontal stress, and the maximum shear stress is concentrated along the edge of the surcharge zone. The vertical stress beneath the surcharge area exhibits a depth-dependent attenuation pattern.

(2) As the surcharge load increases, both the extent and intensity of regions exhibiting large shear-strain increments expand, and the shear band progressively propagates from the slope crest toward the slope toe. The shear-failure zones that form within the slope beneath the edge of the surcharge area and at the slope toe tend to extend toward the mid-slope region.

(3) Under the same surcharge load, zones exhibiting pronounced shear-strain increments are more likely to form closer to the slope-break line. At a given surcharge location, a larger load leads to a greater extent of shear-strain-increment zone propagation toward the slope toe.

(4) The failure mode of accumulation-slope masses under surcharge loading differs from that of homogeneous-slope landslides and those induced by other loading conditions. The formation and evolutionary mechanism of surcharge-triggered landslides proceeds as follows: surcharge application → disturbance of the initial equilibrium → initiation of rear-edge tensile cracking → sliding of the upper soil mass → frontal compression and bulging → propagation and coalescence of the sliding surface → overall sliding failure.

(5) Accumulation slopes exhibit marked failure sensitivity under surcharge loading, and the progressive development of rear-edge tensile cracking, frontal bulging, and deep-seated shear zones should be regarded as key indicators for monitoring.

Due to limitations in the experimental conditions, this study still has several constraints:

(1) The experiment adopted only a static, staged surcharge-loading scheme and did not consider influences such as rapid load application during construction, mechanical vibration, or traffic-induced dynamic loading.(2) To simplify the computations, the numerical model employed the Mohr-Coulomb constitutive relation and did not incorporate mesoscopic effects such as structural-plane weakening or particle breakage, which may cause the simulation results to be somewhat conservative or idealized.

Future research can be further advanced in the following directions:(1) Incorporating high-precision in situ monitoring data to establish a multi-scale validation framework that integrates experimental, numerical, and field observations. (2) Investigating mesoscopic failure mechanisms using discrete-element or particle-flow methods to clarify the fundamental influences of particle gradation and structural characteristics. (3) Introducing dynamic loading and unloading effects to simulate surcharge-induced disturbance responses under realistic construction conditions. (4) Conducting multi-field coupling studies involving surcharge loading, rainfall, seepage, and seismic actions to develop a comprehensive stability-assessment framework for more complex environmental settings.

## Supporting information

**S1 Appendix. A Soil Physical Properties Testing Procedure.**
(DOCX)

## Acknowledgments

We would like to thank the National Key Laboratory of Geohazard Prevention and Geoenvironment Protection, Chengdu University of Technology, for providing experimental equipment support.

## Author contributions

**Conceptualization:** Pei Zuan, Jiali Feng, Fenglin Ren.

**Data curation:** Pei Zuan.

**Investigation:** Pei Zuan, Jiali Feng.

**Software:** Fenglin Ren.

**Supervision:** Pei Zuan.

**Visualization:** Jiali Feng.

**Writing – original draft:** Pei Zuan.

**Writing – review & editing:** Jiali Feng, Fenglin Ren.

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
