## [Decision Letter · Decision Letter 0]

20 Oct 2025

Dear Dr. Feng,

We look forward to receiving your revised manuscript.

Kind regards,

Shamshad Alam, PhD

Academic Editor

PLOS ONE

Journal Requirements:

2. We note that your Data Availability Statement is currently as follows: All relevant data are within the manuscript and its Supporting Information files

3. We notice that your supplementary figures are uploaded with the file type 'Figure'. Please amend the file type to 'Supporting Information'. Please ensure that each Supporting Information file has a legend listed in the manuscript after the references list.

4. Please upload a copy of Figures 1-17, to which you refer in your text on pages. If the figures are no longer to be included as part of the submission please remove all reference to it within the text.

5. Please upload a copy of Supporting Information Tables 1-7 which you refer to in your text on pages 27-28.

Reviewers' comments:

Reviewer's Responses to Questions

**Comments to the Author**

1. Is the manuscript technically sound, and do the data support the conclusions?

Reviewer #1: Yes

Reviewer #2: Yes

2. Has the statistical analysis been performed appropriately and rigorously?

Reviewer #1: Yes

Reviewer #2: Yes

3. Have the authors made all data underlying the findings in their manuscript fully available?

Reviewer #1: Yes

Reviewer #2: Yes

4. Is the manuscript presented in an intelligible fashion and written in standard English?

Reviewer #1: Yes

Reviewer #2: Yes

Reviewer #1: Dear Authors

Manuscript entitled

A landslide study Deformation and failure mechanisms of deposit slope under applied loading

This manuscript) Manuscript Number PONE-D-25-46431( has the potential to be a valuable contribution to the field of engineering geology and slope stability. The core research—integrating physical modeling and numerical analysis to decipher the failure mechanism under external load—is sound and relevant. However, significant revisions are required to address the major concerns regarding methodological clarity, data presentation, and the justification of modeling choices. Once these issues are resolved, the manuscript will be much stronger and suitable for publication.

I recommend that the authors carefully address all points raised in this review, particularly the clarification of the model-prototype scaling and the resolution of the data availability statement in line with journal policy.

Main Comments:

1- the title of the paper is not informative and not clear, so I suggest to the authors to revise the title.

2- The research contributions of the paper should be articulated more clearly. The abstract is not representative of the content and contributions of the paper. The abstract does not seem to properly convey the rigor of research. And also I recommend the author(s) to be more direct to cover the research gap, manuscript's goal, materials and methods, main results, main contributions, main suggestions for future research.

3- The necessity and innovation of the article should be presented to the introduction.

4- Following, you will find some new related references which should be added to literature review:

• https://doi.org/10.22034/jes.2024.1999733.1029

• https://doi.org/10.1016/B978-0-323-89861-4.00041-5

• ttps://doi.org/10.1007/s40996-019-00326-3

5- For readers to quickly catch your contribution, it would be better to highlight major difficulties and challenges, and your original achievements to overcome them, in a clearer way in abstract and introduction.

6- It is suggested to compare the results of the present research with some similar studies which is done before.

7- Much more explanations and interpretations must be added for the Results, that's not enough.

8- Some key parameters are not mentioned. The rationale on the choice of the particular set of parameters should be explained with more details. Have the authors experimented with other sets of values? What are the sensitivities of these parameters on the results?

9- You should add the flow chart in this manuscript for methodology.

10- The conclusion section would be improved by starting with the research goal. The author could present the practical implication of this study. However, further suggestions for future research could be addressed and inserting a paragraph by stating the limitations of this study.

Major Points of Concern

1. Clarity and Coherence of the Experimental and Numerical Setup:

• Scale and Units: There is a critical and confusing discrepancy in the scaling and units between the physical model and the numerical simulation. The physical model uses a geometric scale of 1:200 and applies a maximum load of 2.0 kPa. The numerical simulation, intended to represent the prototype, applies loads of 200-400 kPa. The relationship between these loads is not explained. Is the 2 kPa a model-scale load? If so, what is the corresponding prototype load (it should be 400 kPa, given the geometric scale)? This fundamental relationship must be explicitly stated and justified.

• Numerical Model Justification: The manuscript states that numerical simulation is used to overcome the limitations of the physical model (lines 235-242). However, it then uses a "homogeneous soil mass" for the simulation, whereas the physical model and the field site are clearly composed of multiple strata (Q4del, Q4al+pl, Q3fgl). This simplification undermines the goal of validating the experimental results and representing the prototype. The authors must justify why a homogeneous model was chosen and discuss how this affects the comparability of the results.

2. Data Availability Statement:

• The journal's policy explicitly states that "‘data available on request from the author’ is not sufficient." The current Data Availability Statement (line 471) contradicts this policy. The authors must either:

o Deposit all relevant data in a public repository (e.g., figshare, Zenodo) and provide the accession link/DOI.

o State unequivocally that all data are within the manuscript and its Supporting Information files (which currently do not seem to include the raw data, only figures).

o Provide a compelling, journal-compliant reason for restricting data access.

3. Presentation and Interpretation of Results:

• Figure Referencing: All figures are cited as "S1 Fig," "S2 Fig," etc., in the text, but the corresponding captions and images are embedded at the end of the PDF without formal captions. This makes it extremely difficult to follow the narrative. The figures themselves (e.g., S16_Fig.16.tif) are also very small and lack clear, readable labels and scales. The figures must be properly formatted with high resolution and descriptive captions.

• Quantitative Analysis: The results are largely descriptive. A more quantitative analysis would significantly strengthen the paper. For instance:

o The failure process described in lines 432-437 is a key finding. This should be directly linked to quantitative thresholds from the model (e.g., "rear-edge tensile cracking initiated when the load reached X kPa and the horizontal displacement at sensor X exceeded Y mm").

o The regression equations (Eq 1 and Eq 2) are presented without sufficient context. The R-squared values or other goodness-of-fit metrics are essential to judge their validity. The units for factors A, B, and C in Eq 1 are missing.

4. Discussion and Novelty:

• The discussion section effectively synthesizes the findings but could be strengthened by a more direct comparison with existing literature. The introduction rightly identifies a gap regarding "stress disturbances caused by engineering activities" (line 77). The discussion should explicitly state how the discovered "progressive migration" mechanism (line 435) differs from failure mechanisms triggered by rainfall or earthquakes, as studied by the cited authors (Khan, Gan, Ma, etc.). This would better highlight the manuscript's novel contribution.

Minor Points and Editorial Corrections

• Abstract: The abstract is well-written but could be slightly more specific. For example, "greatest at the slope crest" could be "up to X mm at the slope crest."

• Keywords (line 38): The keywords are run together. They should be separated by semicolons for proper indexing: "External Loading; Deposit slope; Physical Model Experiment; Slope Deformation and Failure".

• Author Contribution (lines 466-468): The formatting is inconsistent. It should be standardized, e.g., Pei Zuan: Conceptualization, Data curation, Writing – original draft, Investigation, Supervision.

• Line 110-115: The stratigraphic nomenclature (Q4[m], Q4pø, Q4[de]) is confusing. The symbol "ø" is unusual, and the use of brackets and superscripts is inconsistent. Please use a consistent format throughout.

• Line 199-204: The description of the model box and loading is good, but the size of the sandbags (1-meter-long) seems to be a prototype-scale description leaking into the model-scale description. Clarify.

• Language and Grammar: The manuscript is generally well-written but would benefit from a thorough proofread by a native English speaker to correct minor grammatical awkwardness (e.g., "Sensor 5 exceeding readings..." in line 300).

Thanks

Reviewer #2: Title: A landslide study Deformation and failure mechanisms of deposit slope under applied loading

The introduction provides a strong global context on medical geology and the role of a sustainable national framework. This work shows the authors' understanding of international best practices. However, the section sometimes lacks organizational flow, with abrupt transitions between the data, technologies, and theoretical models.

The Objectives and Methodology section presents a relevant goal. However, the methodology lacks critical detail—there is no clear explanation of search strategies, databases used, selection processes, or thematic coding. A structured PRISMA-style flowchart or summary table should be added for clarity. Inclusion and exclusion criteria are vaguely addressed, and no justification is given for the types of studies.

The Results and Findings section is well organized. However, it needs to integrate quantitative data (casualties, damages) with qualitative assessments and include useful visuals like photos and maps.

The conceptual framework and research questions are poorly written. Furthermore, the research needs to be attractive.

The data collection and data analysis in this paper are relatively concise, and the author is advised to enrich this part of the content.

General Comment:

The authors may consider making their paper more concise and focusing more on the Landslide and slope Stability component they wish to introduce or highlight. The rationale behind the introduction of this framework needs to be clearly established. In addition, providing a conceptual framework on landslides would enhance appreciation and understanding of the framework.

Introduction:

The authors did not explicitly present the rationale of the study. Some of the information provided is scattered and needs to be stated more logically and cohesively.

It would also strengthen the paper if the objectives of the study were clearly mentioned.

Further explanation of landslides and slope Stability is essential, particularly in demonstrating how it is more encompassing than the "landslide" theme.

Materials and Methods:

This section appears to be an extended well and actual materials or methods used in the study were presented.

Results:

For the statement, “The natural hazards from landslides pose…”, it is suggested to simply use “hazards” since they are no longer natural in origin, in this context.

Under “Remediation Frameworks for landslide prone zones,” no framework was presented. The authors may consider discussing other relevant frameworks and how landslides and slope Stability could address the gaps in those frameworks.

It is also recommended that greater emphasis be placed on sections directly relevant to the proposed framework, while subsections that are less relevant need not be discussed in detail.

Discussion:

The discussion requires further improvement to comprehensively or concisely address the results presented more extensively.

Conclusions:

Ensure that the statements in this section accurately reflect the extent of the research and review conducted by the authors. Avoid broad generalizations.

**Do you want your identity to be public for this peer review?** For information about this choice, including consent withdrawal, please see our Privacy Policy

Reviewer #1: No

Reviewer #2: **Yes: ** Fredrick Kayusi

---

## [Author Response · Author response to Decision Letter 1]

3 Dec 2025

Due to the length and detail of the reviewer and editor comments, we have provided a complete point-by-point response in the separately uploaded response letter. Please refer to the attached detailed reply for our full responses. We remain happy to provide any further clarification if needed.

---

## [Decision Letter · Decision Letter 1]

11 Dec 2025

Mechanisms   of   Deformation   and   Failure   in   C olluvial   Slope   under   artificial   surcharge   loading

PONE-D-25-46431R1

Dear Dr. Feng,

We’re pleased to inform you that your manuscript has been judged scientifically suitable for publication and will be formally accepted for publication once it meets all outstanding technical requirements.

Kind regards,

Shamshad Alam, PhD

Academic Editor

PLOS One

Additional Editor Comments (optional):

Reviewers' comments:

Reviewer's Responses to Questions

**Comments to the Author**

Reviewer #1: All comments have been addressed

Reviewer #2: All comments have been addressed

2. Is the manuscript technically sound, and do the data support the conclusions?

Reviewer #1: Yes

Reviewer #2: Yes

3. Has the statistical analysis been performed appropriately and rigorously?

Reviewer #1: Yes

Reviewer #2: Yes

4. Have the authors made all data underlying the findings in their manuscript fully available?

Reviewer #1: Yes

Reviewer #2: Yes

5. Is the manuscript presented in an intelligible fashion and written in standard English?

Reviewer #1: Yes

Reviewer #2: Yes

Reviewer #1: Dear Authors

Although some of my comments were not fully addressed, it is evident that the authors have made a sincere effort to improve the manuscript. In particular, the methodology and interpretation of the model have been significantly expanded and strengthened. Therefore, I find the revised version acceptable for publication.

Thank you.

Reviewer #2: I have reviewed the revised manuscript and I ascertain that the authors have addressed all the issues I raised to streamline the article. Thanks.

**Do you want your identity to be public for this peer review?** For information about this choice, including consent withdrawal, please see our Privacy Policy

Reviewer #1: **Yes: ** Ebrahim Sharifi Teshnizi

Reviewer #2: **Yes: ** Fredrick Kayusi

---

## [Editor Report · Acceptance letter]

PONE-D-25-46431R1

PLOS One

Dear Dr. Feng,

I'm pleased to inform you that your manuscript has been deemed suitable for publication in PLOS One. Congratulations! Your manuscript is now being handed over to our production team.

Kind regards,

on behalf of

Dr. Shamshad Alam

Academic Editor

PLOS One